# Tomosyn affects dense core vesicle composition but not exocytosis in mammalian neurons

Aygul Subkhangulova[1]*, Miguel A Gonzalez-Lozano[2†], Alexander JA Groffen[3], Jan RT van Weering[3], August B Smit[2], Ruud F Toonen[1], Matthijs Verhage[1,3]*

[1]Department of Functional Genomics, Center for Neurogenomics and Cognitive Research (CNCR), Vrije Universiteit (VU) Amsterdam, Amsterdam, Netherlands; [2]Department of Molecular and Cellular Neurobiology, Center for Neurogenomics and Cognitive Research (CNCR), Vrije Universiteit (VU) Amsterdam, Amsterdam, Netherlands; [3]Department of Human Genetics, Center for Neurogenomics and Cognitive Research (CNCR), Amsterdam University Medical Center (UMC), Amsterdam, Netherlands

*For correspondence:
a.subkhangulova@vu.nl (AS);
matthijs@cncr.vu.nl (MV)

Present address: †Department of Cell Biology, Harvard Medical School, Boston, United States

Competing interest: The authors declare that no competing interests exist.

**Abstract** Tomosyn is a large, non-canonical SNARE protein proposed to act as an inhibitor of SNARE complex formation in the exocytosis of secretory vesicles. In the brain, tomosyn inhibits the fusion of synaptic vesicles (SVs), whereas its role in the fusion of neuropeptide-containing dense core vesicles (DCVs) is unknown. Here, we addressed this question using a new mouse model with a conditional deletion of tomosyn (*Stxbp5*) and its paralogue tomosyn-2 (*Stxbp5l*). We monitored DCV exocytosis at single vesicle resolution in tomosyn-deficient primary neurons using a validated pHluorin-based assay. Surprisingly, loss of tomosyns did not affect the number of DCV fusion events but resulted in a strong reduction of intracellular levels of DCV cargos, such as neuropeptide Y (NPY) and brain-derived neurotrophic factor (BDNF). BDNF levels were largely restored by re-expression of tomosyn but not by inhibition of lysosomal proteolysis. Tomosyn's SNARE domain was dispensable for the rescue. The size of the trans-Golgi network and DCVs was decreased, and the speed of DCV cargo flux through Golgi was increased in tomosyn-deficient neurons, suggesting a role for tomosyns in DCV biogenesis. Additionally, tomosyn-deficient neurons showed impaired mRNA expression of some DCV cargos, which was not restored by re-expression of tomosyn and was also observed in Cre-expressing wild-type neurons not carrying loxP sites, suggesting a direct effect of Cre recombinase on neuronal transcription. Taken together, our findings argue against an inhibitory role of tomosyns in neuronal DCV exocytosis and suggests an evolutionary conserved function of tomosyns in the packaging of secretory cargo at the Golgi.

## Editor's evaluation

This study examines the function of Tomosyn, in dense core vesicle fusion using CRE-mediated deletion in neuronal cultures from mice expressing conditional alleles of tomosyn and tomosyn-2. Tomosyn is a large soluble SNARE protein, where earlier work in multiple species suggested that it functions as a competitive inhibitor of cognate SNARE interactions impairing fusion. The authors show that while loss of tomosyns did not affect dense core vesicle exocytosis, it reduced the expression of several key dense core cargos, including BDNF, with limited impact on intracellular vesicle trafficking or Golgi function. These results suggest that tomosyns impact neuropeptide and neurotrophin secretion by regulating dense core vesicle cargo production but not exocytosis.

## Introduction

Brain activity relies on the precisely regulated secretion of different chemical messengers, and most neurons co-release multiple neurotransmitters (*Nusbaum et al., 2017*). Classical small-molecule neurotransmitters (e.g. glutamate, γ-aminobutyric acid) are released from SVs, whereas neuropeptides and neurotrophic factors are packaged in and secreted from DCVs. SV and DCV substantially differ in their morphology and mechanisms of formation (reviewed in *De Camilli and Jahn, 1990*; *Gondré-Lewis et al., 2012*). While SVs are loaded with their content directly at synapses, where they subsequently fuse with the presynaptic membrane, DCVs are formed in neuronal somas at the trans-Golgi network (TGN), from where they travel to the release sites that are not confined to synapses. Despite these and many other differences between the two main types of neuronal secretory vesicles, $Ca^{2+}$-triggered exocytosis of both SV and DCV in most neurons requires an identical basic machinery consisting of SNARE proteins: syntaxin-1 and SNAP25 at the plasma membrane, and VAMP2 on the vesicle (*Südhof, 2013*; *Shimojo et al., 2015*; *Arora et al., 2017*; *Hoogstraaten et al., 2020*). The SNARE domains of these proteins form a tight complex that brings the two membranes together and drives vesicle fusion. In addition to the basic SNARE machinery, many other proteins participate in the fusion process (reviewed in *Südhof, 2013*; *Rizo and Xu, 2015*), of which some are differentially required for SV and DCV exocytosis (*Persoon et al., 2019*; *Moro et al., 2021*). The differences in molecular machinery may explain why SV and DCV exocytosis are triggered upon different stimulation paradigms and at different locations within neurons.

Tomosyn (STXBP5) is a large, evolutionary conserved SNARE protein discovered as an interactor of syntaxin-1 in rat brain (*Fujita et al., 1998*; *Masuda et al., 1998*). Tomosyn is expressed in a wide range of specialized secretory cell types, such as neurons, platelets, and neuroendocrine cells, suggesting a fundamental role of tomosyn in regulated secretion. Mammals express a second tomosyn gene, *STXBP5L*, which encodes tomosyn-2 with a nearly 80% amino acid similarity (*Groffen et al., 2005*). The two genes have a partially overlapping expression pattern in the mouse brain suggesting functional redundancy (*Groffen et al., 2005*; *Barak et al., 2010*). Mutations in both genes were found in patients with neurodevelopmental disorders, suggesting a role of tomosyns in human brain function (*Matsunami et al., 2013*; *Cukier et al., 2014*; *De Rubeis et al., 2014*; *Kumar et al., 2015*).

Tomosyn is comprised of a C-terminal SNARE domain and N-terminal WD40 repeats folded into two seven-bladed β-propellers. The SNARE-domain of tomosyn is homologous to the SNARE domain of VAMP2 and mediates binding to syntaxin-1 and SNAP25. The resulting complex is remarkably similar in structure and stability to the fusogenic complex of syntaxin-1/SNAP25 with VAMP2 (*Pobbati et al., 2004*). However, in contrast to VAMP2 and most other SNARE proteins, tomosyn does not contain a transmembrane anchor and, therefore, cannot drive membrane fusion. This feature led to a hypothesis that tomosyn functions as an inhibitor of fusion by competing with VAMP2 for binding to syntaxin-1/SNAP25 (*Hatsuzawa et al., 2003*). In support of this hypothesis, overexpression of tomosyn in neuroendocrine PC12 cells and adrenal chromaffin cells inhibits the fusion of secretory vesicles (*Fujita et al., 1998*; *Hatsuzawa et al., 2003*; *Yizhar et al., 2004*), and, conversely, loss of tomosyn in the nematode, fruit fly and mouse brain increases SV release probability and enhances synaptic transmission (*Gracheva et al., 2006*; *McEwen et al., 2006*; *Chen et al., 2011*; *Sauvola et al., 2021*; *Sakisaka et al., 2008*; *Ben-Simon et al., 2015*). In addition, loss of tomosyn in *C. elegans* augments neuropeptide secretion, as evidenced by decreased DCV abundance at synapses and increased accumulation of a DCV cargo in scavenger cells (*Gracheva et al., 2007*; *Ch'ng et al., 2008*). Together, these studies suggest an evolutionary conserved role of tomosyn as an inhibitor of secretory vesicle exocytosis.

However, several lines of evidence argue against this conclusion. First, rat sympathetic neurons show a decrease in neurotransmitter release from SV upon depletion of tomosyn (*Baba et al., 2005*). Second, platelets from *Stxbp5*-null mice show severely reduced exocytosis of all types of secretory granules, which results in excessive bleeding and impaired thrombus formation in vivo (*Zhu et al., 2014*; *Ye et al., 2014*). Finally, the loss of SRO7 and SRO77, two yeast orthologs of mammalian tomosyns, causes a strong defect in the exocytosis of post-Golgi vesicles (*Lehman et al., 1999*). Collectively, these studies suggest a positive role of tomosyn and its yeast orthologs in exocytosis. Thus, despite the clear importance of tomosyn in regulated secretion, its role in this process remains controversial.

In the brain, the role of mammalian tomosyn has been studied with a focus on SV exocytosis, whereas its effect on DCV exocytosis is unknown. In this study, we addressed this question using a

mouse model with the inducible loss of all tomosyn and tomosyn-2 isoforms and show that tomo-syns do not affect neuronal DCV exocytosis but, paradoxically, regulate intracellular levels of over-expressed and endogenous secretory cargos. Thus, tomosyns differentially modulate two regulated secretory pathways in neurons (secretion from SVs and DCVs) and have an additional role in the regulation of DCV composition.

## Results

To explore the role of tomosyn in neuronal secretion, we generated a new mouse model with a conditional deletion of tomosyn (*Stxbp5*) and tomosyn-2 (*Stxbp5l*) (*Figure 1—figure supplement 1A*). Primary hippocampal neurons isolated from these mice showed a complete loss of tomosyns expression when transduced with a lentivirus encoding *Cre*-recombinase, as shown by Western blot (WB) and immunocytochemistry (ICC) (*Figure 1—figure supplement 1B–C*). As a control, we used neurons isolated from the same litter but expressing a defective *Cre*-recombinase that lacks the catalytic domain (ΔCre). Double knockout (DKO) neurons developed normally in culture and did not show any defects in neurite outgrowth and synapse formation as evidenced by normal dendrite length and density of synaptophysin puncta (*Figure 1—figure supplement 1D–G*).

To examine DCV fusion, we used a modification of a validated reporter consisting of NPY fused to super-ecliptic pHluorin (*Figure 1A*, *Figure 1—figure supplement 2*; *Persoon et al., 2018*; *Nassal et al., 2022*). When expressed in primary neurons, this reporter localized almost exclusively to DCVs, as indicated by its co-localization with endogenous DCV markers IA-2 and chromogranins (CHGA, CHGB) (*Figure 1—figure supplement 2*). Fusion of labeled DCVs with the plasma membrane was stimulated by repetitive bursts of action potentials delivered at 50 Hz to single neurons grown on glial micro-islands. Individual fusion events were detected as transient increases in fluorescence intensity along neurites, as exemplified in *Figure 1B*. Our stimulation paradigm resulted in a steep accumulation of fusion events that plateaued shortly before the end of the stimulation (*Figure 1C*). Most of these events (~80%) were previously shown to be of axonal origin (*Persoon et al., 2018*). Virtually no fusion events were detected in neurons of both genotypes in the absence of electric stimulation (first 30 s of recordings or in the period after stimulation). Surprisingly, the total number of evoked fusion events was similar between control and DKO neurons, with a tendency (p=0.09) towards a decrease in DKO neurons (*Figure 1D*). The intracellular pool of the DCV reporter, visualized by a short application of 50 mM $NH_4Cl$ to neurons to dequench intravesicular NPY-pHluorin, was decreased by 28% in DKO neurons (*Figure 1E*). The released fraction, i.e., the number of fusion events normalized to the intracellular pool of vesicles, was nearly identical between the genotypes and comprised approximately 8% of the intracellular content, consistent with the results of our previous studies (*Persoon et al., 2018*; *Moro et al., 2020*; *Figure 1F*). Also, the averaged fluorescence peak of individual fusion events, an estimate of NPY loading per vesicle, was not affected by the loss of tomosyns (*Figure 1G*). Thus, tomosyns are dispensable for the evoked fusion of neuronal DCVs in hippocampal neurons.

Tomosyn was proposed to act as an inhibitor of vesicle fusion by competing with VAMP2 for the binding to syntaxin-1 and SNAP25. Hence, loss of tomosyns may result in the increased formation of ternary synaptic SNARE complexes consisting of syntaxin-1, SNAP25, and VAMP2. These complexes can be detected based on their resistance to SDS, as was shown for tomosyn-deficient fly brains (*Sauvola et al., 2021*). We examined levels of the assembled complexes in DKO neurons by SDS-PAGE on non-boiled neuronal lysates, since SDS-resistant complexes disassemble at temperatures higher than 60 °C (*Hayashi et al., 1994*; *Otto et al., 1997*; *Figure 1H*). As expected, a band of approximately 80 kDa that corresponds to the ternary complexes was readily detected with a syntaxin-1 antibody in non-boiled lysates and absent in lysates heated at 95 °C. This band shifted in size upon expression of tetanus toxin light chain (TeNT), which results in cleavage of VAMP2, thus confirming that the detected complexes contain VAMP2 (*Figure 1—figure supplement 3C*). No difference in the intensity of the 80 kDa band was observed between control and DKO cultures. The same results were obtained when non-boiled lysates were probed with antibodies against SNAP25 and VAMP2 (*Figure 1—figure supplement 3A–B*). These data suggest that tomosyns do not interfere with the bulk formation of syntaxin-1 containing synaptic SNARE complexes in cultured neurons. Interestingly, in addition to the 80 kDa band, non-boiled WT lysates showed a band of approximately 200 kDa, detected with antibodies against syntaxin-1 and SNAP25. This band was absent in DKO lysates and, therefore, likely represents a complex of tomosyn with syntaxin-1 and SNAP25, suggesting that this

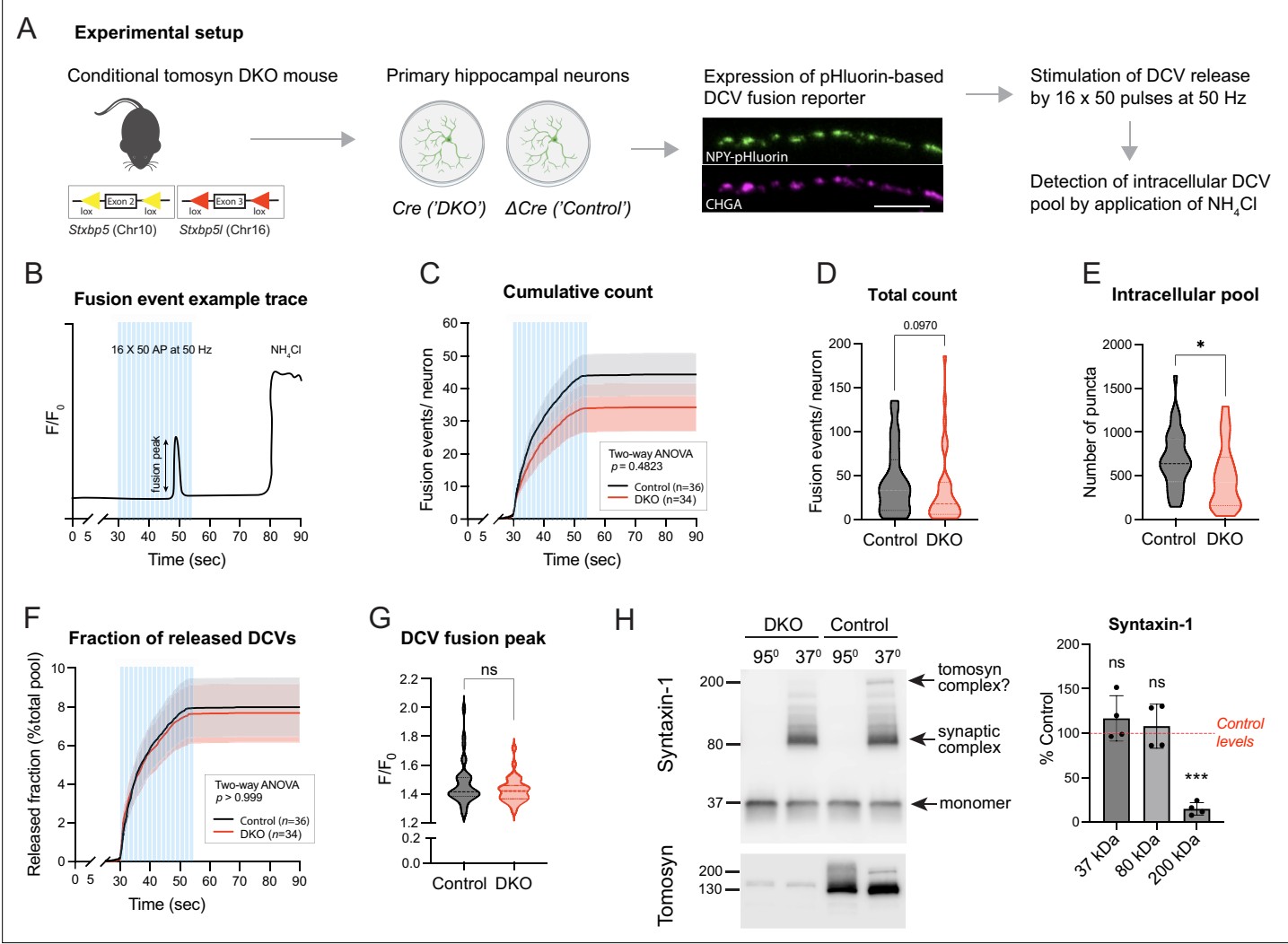

**Figure 1.** Loss of tomosyns does not affect the fusion of dense core vesicles (DCVs) in mouse hippocampal neurons. (**A**) Experimental strategy for measuring DCV fusion in single isolated neurons. (**B**) Exemplified fluorescence intensity trace of a neuronal DCV fusion event evoked by electric stimulation (shown as blue bars). Stimulation paradigm consisted of 16 bursts of 50 pulses ('AP') delivered at 50 Hz. Transient increase in relative fluorescence intensity ($F/F_0$) indicates a fusion event during the stimulation. At the end of a recording session, neurons were flushed with 50 mM $NH_4Cl$ to visualize not fused DCVs. (**C**) Cumulative count of fusion events detected per single neuron. Data are shown as mean ± SEM and were analyzed by two-way ANOVA. n=34–36 neurons/genotype from three culture preparations. (**D**) Total count of fusion events detected per single neuron. Data are shown as violin plots with median (dashed line) and quartiles (dotted lines) and analyzed using the Mann-Whitney test. (**E**) Intracellular pool of pHluorin-labeled DCVs determined by application of 50 mM $NH_4Cl$ after the stimulation. Data are shown as violin plots with median (dashed line) and quartiles (dotted lines) and analyzed using a two-tailed unpaired *t*-test. *p<0.05. (**F**) Cumulative count of fusion events normalized to the intracellular pool of DCVs (released fraction). Data are shown as mean ± SEM and were analyzed by two-way ANOVA. (**G**) Peak in fluorescence intensity that results from a single fusion event (averaged per neuron). Data are shown as violin plots with median (dashed line) and quartiles (dotted lines) and analyzed using the Mann-Whitney test. ns: not significant. (**H**) Levels of syntaxin-1 containing SDS-resistant 80 kDa SNARE complexes are unchanged in Double knockout (DKO) neurons. Boiled and non-boiled lysates were subjected to SDS-PAGE and membranes were probed with an antibody against syntaxin-1 (STX1). Temperature-sensitive complexes were detected at 80 kDa (synaptic SNARE complex) and 200 kDa (possibly complex of syntaxin-1 and SNAP25 with tomosyn). Smear at >80 kDa represents complex multimers. Band intensities in DKO were normalized to the control, plotted as mean ± SD, and analyzed using a one-sample *t*-test. Dots represent independent culture preparations (n=4). ns: not significant; ***p<0.001.

The online version of this article includes the following source data and figure supplement(s) for figure 1:

**Source data 1.** Uncropped western blot (WB) images for *Figure 1*.

**Figure supplement 1.** Validation of the mouse model with the conditional deletion of tomosyn and tomosyn-2.

**Figure supplement 1—source data 1.** Uncropped western blot (WB) images for *Figure 1—figure supplement 1B*.

**Figure supplement 2.** Design and validation of the pHluorin-based dense core vesicle (DCV) fusion reporter.

*Figure 1 continued on next page*

Figure 1 continued

**Figure supplement 3.** Levels of SNAP25- and VAMP2-containing SDS-resistant SNARE complexes are unchanged in double knockout (DKO) neurons.

**Figure supplement 3—source data 1.** Uncropped western blot (WB) images for *Figure 1—figure supplement 3*.

complex is resistant to SDS in vivo. This 200 kDa complex was also detected in control neurons with an antibody against tomosyn (*Figure 1H*).

Despite normal DCV exocytosis, intracellular levels of the DCV reporter were decreased in DKO neurons (*Figure 1E*). This decrease was also observed in tetrodotoxin (TTX)-silenced network cultures of DKO neurons that were not subjected to electric stimulation (*Figure 2A–B*). Activity of the synapsin promoter used to drive expression of the DCV reporter was not significantly altered in DKO neurons, as shown by normal expression of enhanced green fluorescent protein (EGFP) from the same promoter (*Figure 2—figure supplement 1*). Re-expression of the most ubiquitous tomosyn isoform (m-isoform of *Stxbp5*) restored levels of the overexpressed NPY-pHluorin in DKO neurons (*Figure 2A–B*). These data suggest that tomosyns affect the amount of the overexpressed DCV cargo and/or the number of DCVs in hippocampal neurons.

We next examined if endogenous DCV cargos are also affected by the loss of tomosyns. Levels of BDNF, one of the most common DCV cargos in excitatory hippocampal neurons (*Dieni et al., 2012*), were decreased by more than half in DKO neurons as shown by ICC and WB analysis (*Figure 2C–F*). Levels of IA-2 (PTPRN), a transmembrane DCV-resident protein commonly used as a DCV marker, were also lower, by 25%, in DKO neurons as indicated by ICC, and by approximately 50% as evidenced by WB analysis, which allows to distinguish a proteolytically processed (mature) form of IA-2 that runs at approximately 65 kDa (*Hermel et al., 1999*; *Figure 2—figure supplement 2*). The cellular distribution of the two DCV markers was similar between the genotypes, with a comparable reduction in somas and neurites of DKO neurons and no re-distribution between the two compartments as detected by ICC. Markers of lysosomes/late endosomes, LAMP1, and LIMP2, were not affected by the loss of tomosyns, as shown by ICC and WB (*Figure 2—figure supplement 3*). Re-expression of tomosyn (isoform 1 m, NM_001081344) largely restored BDNF levels in DKO neurons (to approximately 80% of WT levels), as shown by ICC and WB (*Figure 2C–D and G–H*). The SNARE-domain of tomosyn was dispensable for the partial rescue of BDNF levels, since expression of a truncated tomosyn mutant lacking the SNARE-domain (ΔSN) rescued BDNF levels in DKO to the same extent as re-expression of the full-length (FL) tomosyn (*Figure 2G–H*). Taken together, these data suggest that tomosyns affect intracellular levels of some endogenous DCV cargos in a SNARE-domain independent manner.

Levels of BDNF and other secretory cargos are known to be regulated by lysosomal degradation. We, therefore, tested if an increase in BDNF degradation could account for its decreased levels in the absence of tomosyns. Inhibition of lysosomal proteases by leupeptin for 24 hr resulted in a twofold increase of BDNF levels in neurons of both genotypes but failed to raise BDNF expression in DKO to WT levels (*Figure 2G and I*), indicating that increased cargo degradation does not explain the decreased BDNF levels in DKO neurons.

We next asked whether levels of any other (secretory) proteins are affected by the loss of tomosyns. To address this question in an unbiased manner, we compared the proteome of cultured control and DKO neurons by mass spectrometry. We detected approximately 8500 proteins, of which around 3% were differentially expressed ($log_2$ fold change threshold 0.3) (*Figure 3A*). Gene ontology (GO) analysis of proteins decreased in DKO neurons indicating a strong enrichment of proteins related to the cytoskeleton and secretory granules (*Figure 3B*). As expected, both tomosyns were not detected in DKO neurons (*Figure 3C*). Interestingly, BDNF was also barely detected in DKO (only in one out of six biological replicates), while readily detected across all replicates in control conditions (*Figure 3C*), confirming a strong decrease in BDNF levels in DKO neurons. Many common DCV cargos, such as granins VGF and SCG2, cargo processing enzymes PCSK1 and CPE, and transmembrane DCV proteins IA-2 (PTPRN) and PAM were also decreased, albeit to a different extent, in DKO (*Figure 3D*). Most of the up-regulated proteins were components of the mitochondrial electron transport chain, suggesting that an increase in oxidative phosphorylation is a metabolic feature of DKO neurons (*Figure 3B and D*, *Supplementary file 1*). In contrast to the proteins regulating oxidative phosphorylation, most other mitochondrial proteins, for example, proteins responsible for mitochondrial DNA replication and transcription, were unchanged in DKO. A selected number of synaptic proteins were affected in DKO neurons but, in contrast to the DCV-resident proteins, SV proteins did not follow any general

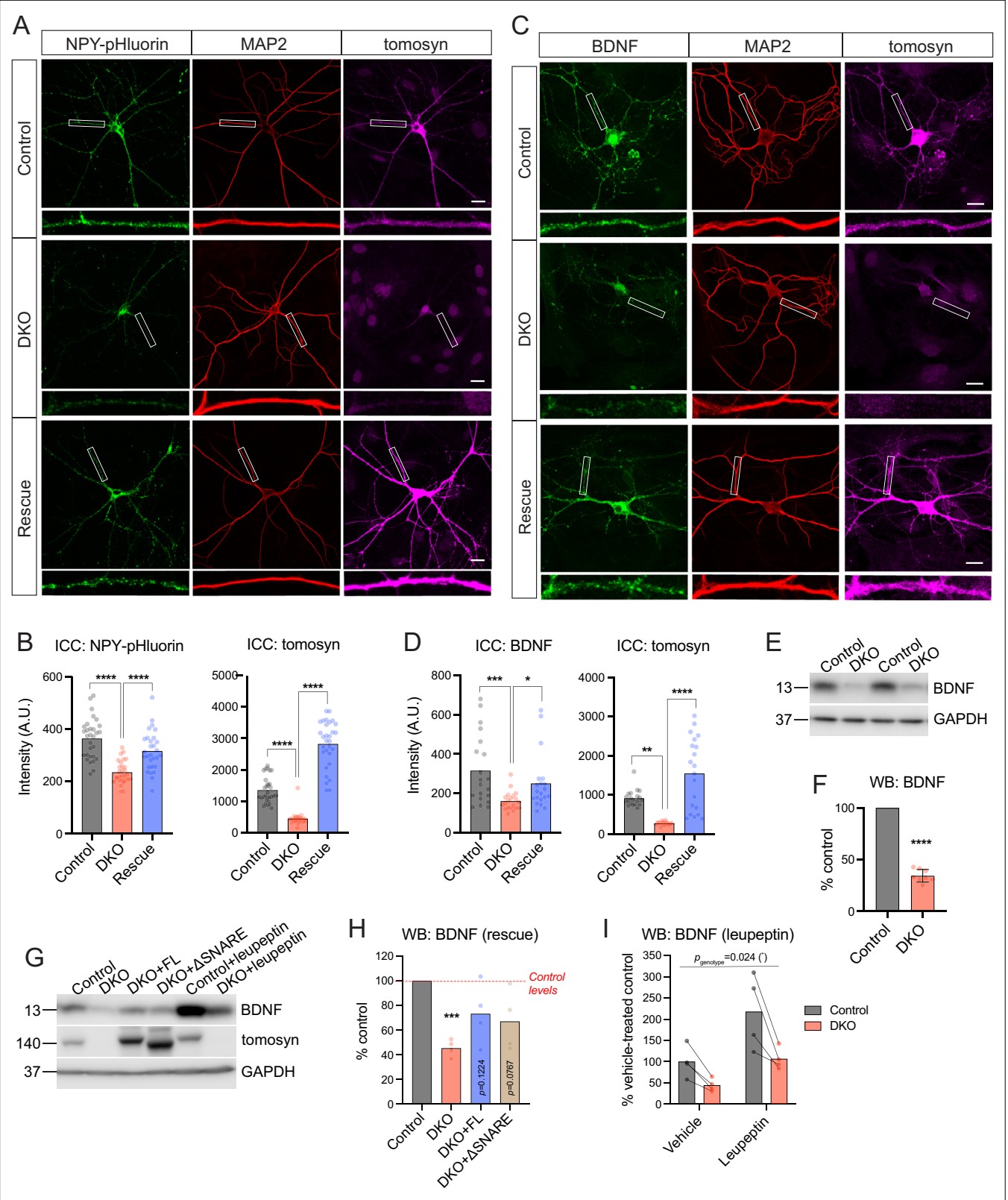

**Figure 2.** Tomosyns regulate the number and/or composition of neuronal dense core vesicles (DCVs). (**A**) Representative images of the DCV reporter (NPY-pHluorin) in control, double knockout (DKO), and DKO re-expressing tomosyn (*Stxbp5* isoform m, 'Rescue') neurons. Neurons were grown in mass cultures, silenced with sodium channel blocker tetrodotoxin (TTX, 1 μM) for 48 hr, and fixed on DIV14. White boxes indicate zoomed-in segments of neurites shown under every image. Scale bar 20 μm. (**B**) Mean intensity of NPY-pHluorin and tomosyn immunostaining in control, DKO, and rescued

*Figure 2 continued on next page*

*Figure 2 continued*

neurons exemplified in (**A**). Data were analyzed using one-way ANOVA with Tukey's multiple comparisons post hoc test. n=29–30 neurons (plotted as dots)/ genotype. ****p<0.0001. (**C**) Representative images of BDNF immunostaining in control, DKO, and DKO re-expressing tomosyn neurons. Neurons were grown on glial microislands and fixed on DIV16. White boxes indicate zoomed-in segments of neurites shown under every image. Scale bar 20 μm. (**D**) Quantification of the mean BDNF and tomosyn intensity in control, DKO and rescued neurons exemplified in (**C**). Data were analyzed using the Kruskal-Wallis test with Dunn's multiple comparisons post hoc test (BDNF) and one-way ANOVA with Tukey's multiple comparisons post hoc test (tomosyn). n=20 neurons (plotted as dots)/ genotype. ****p<0.0001. (**E**) BDNF levels as detected by western blot (WB) in lysates of control and DKO neuronal cultures. Equal loading was verified by immunodetection of GAPDH. (**F**) Quantification of the BDNF band intensity from WB exemplified in (**E**). BDNF levels in DKO neurons were normalized to control levels in the corresponding culture. DKO data are plotted as mean ± SD and were analyzed using one-sample *t*-test. n=8 samples/genotype from four culture preparations. ****p<0.0001. (**G**) Re-expression of tomosyn (either full length, 'FL,' or a truncated mutant lacking the SNARE domain, 'ΔSNARE') partially restores BDNF levels in DKO neurons as detected by WB. Same WB shows that an inhibition of lysosomal proteolysis by leupeptin (50 μM for 24 hr) does not equalize BDNF levels between control and DKO neurons. Immunodetection of tomosyn was used to validate the expression and the correct size of the rescue constructs. Equal loading was verified by immunodetection of GAPDH. (**H**) Quantification of the BDNF band intensity from WB exemplified in (**G**). BDNF levels in DKO neurons are shown as the mean % of control levels. Data were analyzed using one-sample *t*-test comparing to 100% (control levels). n=4 samples/genotype from two culture preparations. ***p<0.001. (**I**) Quantification of the BDNF band intensity in leupeptin-treated samples from WB exemplified in (**G**). BDNF levels in all groups are shown as % of the averaged vehicle-treated control levels. Bars indicate mean values. Data were analyzed using a two-way repeated measures ANOVA. n=4 samples/genotype from two culture preparations.

The online version of this article includes the following source data and figure supplement(s) for figure 2:

**Source data 1.** Uncropped western blot (WB) images for *Figure 2E and G*.

**Figure supplement 1.** Expression of enhanced green fluorescent protein (EGFP) under the control of synapsin promoter is not affected in double knockout (DKO) neurons.

**Figure supplement 2.** Levels of a transmembrane dense core vesicle (DCV) marker, IA-2 (PTPRN), are decreased in double knockout (DKO) neurons.

**Figure supplement 2—source data 1.** Uncropped western blot (WB) images for *Figure 2—figure supplement 2C*.

**Figure supplement 3.** Loss of tomosyns does not affect levels of endo-lysosomal proteins.

**Figure supplement 3—source data 1.** Uncropped western blot (WB) images for *Figure 2—figure supplement 2C*.

trend in DKO neurons: while some proteins (glutamate transporter VGLUT2, glutamate decarboxylase GAD67) were strongly increased in DKO, others (e.g. SYT12 and SV2B) were reduced to a similar extent, and many (e.g. main glutamate transporter VGLUT1, vesicular GABA transporter VGAT) were not significantly altered (*Figure 3D*). In conclusion, *Cre*-mediated loss of both tomosyns leads to selective changes in neuronal proteome with a reduction in cytoskeletal and DCV cargo proteins, an increase in mitochondrial proteins driving oxidative respiration, and a re-arrangement of synaptic proteome.

Next, we examined the ultrastructural morphology of DCVs in DKO and control neurons by electron microscopy (*Figure 4*). We found DCVs with normal dense cores sparsely distributed among neurites in neurons of both genotypes. Most presynaptic sections did not contain any DCVs, in line with previous reports (*Sorra et al., 2006*; *Persoon et al., 2018*). Percent of synaptic sections containing one or several DCVs was decreased in DKO by 30% (12% of all synaptic sections in DKO versus 17% in control), indicating lowered DCV abundancy in DKO. In addition, DCVs in DKO neurons were slightly reduced in diameter (average 65.01±2.08 nm in DKO versus 71.52±1.62 nm in control), with a high proportion of DKO DCVs being only 40–50 nm in diameter (16% of all DCVs in DKO versus 4% in control) (*Figure 4A–B*). Such a reduction in diameter translates to an approximately 25% decrease in volume. In contrast, the diameter of SV was not affected by the loss of tomosyns (*Figure 4D–E*). Other parameters of synaptic ultrastructure, such as SV number per synapse, length of the active zone, and length of postsynaptic density, were also unchanged in DKO neurons (*Figure 4D and F–H*) and DKO somata showed no obvious abnormalities in organelle morphology (*Figure 4D*). Hence, loss of tomosyns leads to a reduction in DCV size and abundancy, without affecting ultrastructure and the number of SVs.

The fact that the abundancy, composition, and size of neuronal DCVs are affected in DKO neurons suggests that tomosyns may regulate DCV biogenesis. DCVs are formed at the TGN in a sequence of membrane fusion/fission events ensuring correct packaging and processing of DCV cargos. We found that tomosyn was enriched at the TGN of cultured hippocampal neurons, as evidenced by the partial colocalization with a TGN marker, TGN38 (*Figure 5—figure supplement 1A–B*). As reported previously, tomosyn also colocalized with mature DCVs and showed an enrichment at the synapses

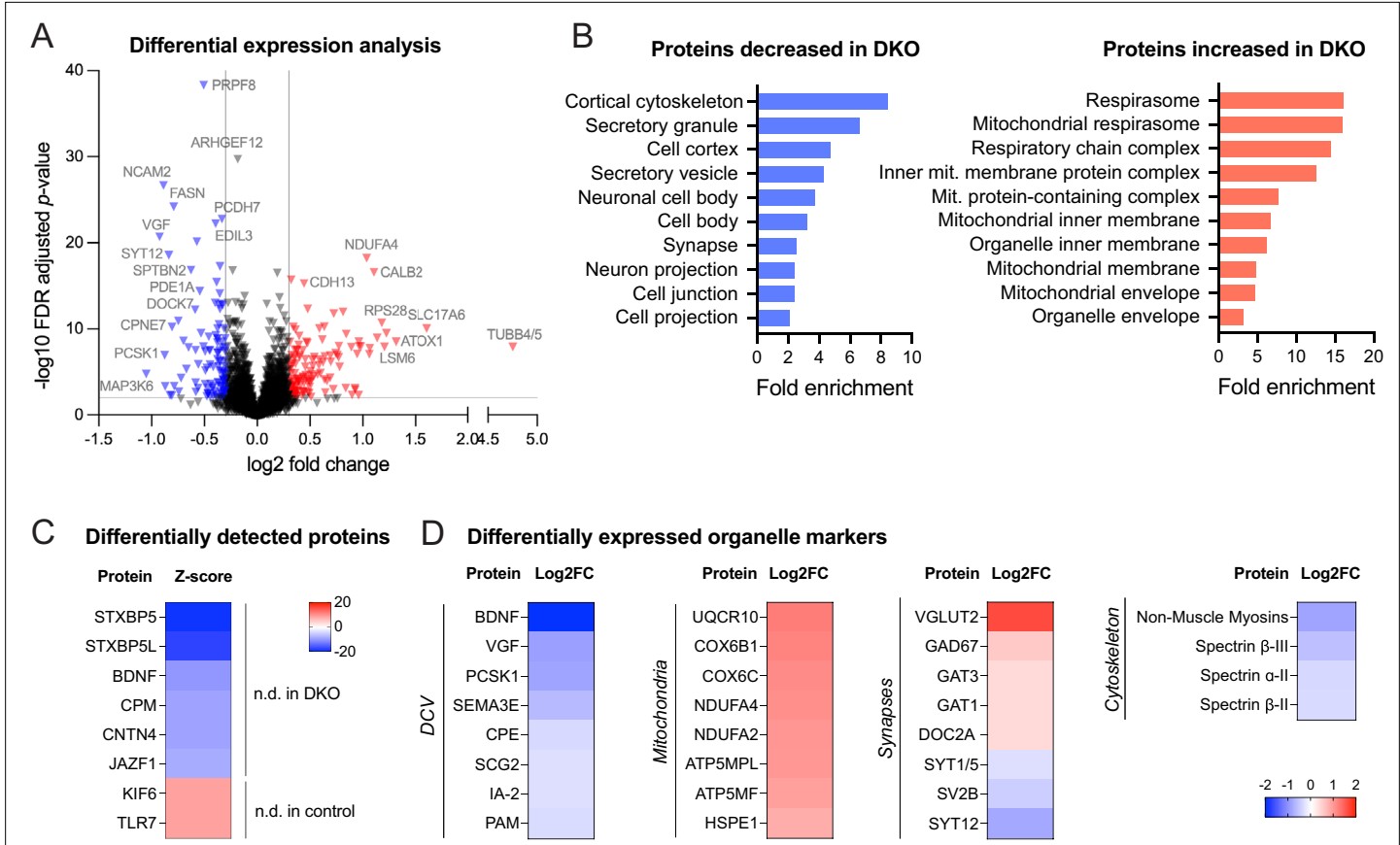

**Figure 3.** Proteomic analysis shows the downregulation of dense core vesicle (DCV) and cytoskeletal proteins in double knockout (DKO) neurons. (**A**) Volcano plot showing results of the differential expression analysis of control and DKO proteome. Gray lines indicate chosen thresholds for log2 fold change (0.3) and false discovery (FDR) adjusted *p*-value (0.01). Proteins decreased or increased in DKO are shown in blue and red, respectively. (**B**) Enriched gene ontology (GO cellular component) terms of the proteins significantly affected in DKO neurons as analyzed by ShinyGO v0.75 (http://bioinformatics.sdstate.edu/go/) with the FDR-adjusted *p*-value cutoff ≤0.05. (**C**) List of proteins reliably detected in one of the genotypes only and, therefore, not included in the differential expression analysis shown in (**A**). Proteins are sorted by the z-score, which reflects the total number of peptides detected per genotype. (**D**) Examples of proteins that are significantly affected in DKO neurons grouped by subcellular compartment. Heat maps show the degree of down- or upregulation in the DKO.

The online version of this article includes the following source data for figure 3:

**Source data 1.** Proteome analysis of neuronal cultures by mass spectrometry - complete list of proteins.

(*Figure 5—figure supplement 1C–F*; *Geerts et al., 2017*), although colocalization of tomosyn with the DCV marker IA-2 was less extensive than with the synaptic marker synaptophysin, possibly due to the fact that IA-2 is also present in endosomes (*Solimena et al., 1996*). The TGN area, visualized by immunostaining of the TGN marker TMEM87A, was slightly decreased in DKO neurons as compared to controls, suggesting a functional importance of tomosyn at the Golgi (*Figure 5A–B*).

We next tested whether tomosyn regulates the packaging and export of a neuropeptide cargo from the TGN using a modification of the Retention Using the Selective Hooks (RUSH) method (*Boncompain et al., 2012*). In this method, streptavidin-binding peptide (SBP)-tagged secretory cargo is retained in the endoplasmic reticulum (ER) until the addition of biotin due to the expression of the ER-localized streptavidin. Biotin disrupts SBP/streptavidin interaction and induces a synchronized transport of the cargo from the ER along the secretory pathway (*Figure 5C–D*). As a neuropeptide cargo, we used NPY fused to SBP and EGFP, as described previously (*Emperador-Melero et al., 2018*). In the absence of biotin, NPY-EGFP-SBP was diffusely distributed along cell soma and processes, in agreement with its localization to the ER. Addition of biotin induced a wave of NPY-EGFP-SBP accumulating in the Golgi first, and then leaving the Golgi as discrete puncta (*Figure 5E*, *Figure 5—video 1*). These puncta co-localized with an endogenous DCV marker, CHGA, indicating correct packaging of the NPY

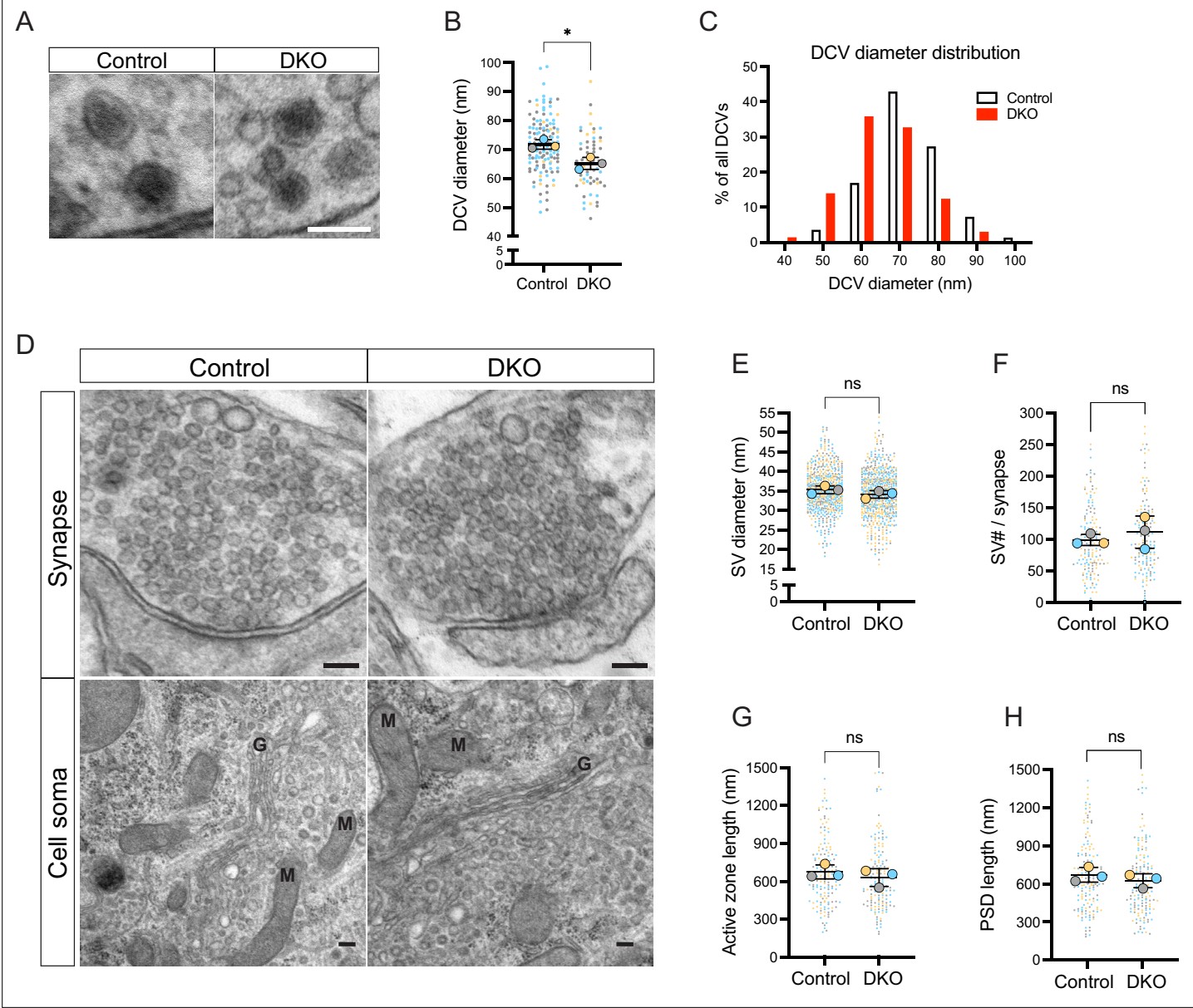

**Figure 4.** Loss of tomosyns results in a decrease of dense core vesicle (DCV) but not of synaptic vesicle (SV) size. (**A**) Example EM images show the normal appearance of DCVs with a typical dense core in double knockout (DKO) neurons. Scale bar 100 nm. (**B**) DCV diameter is reduced in DKO neurons, as quantified from EM images exemplified in (**A**). Data are shown as a SuperPlot, where averages from three independent cultures (biological replicates) are shown as large circles and single observations (DCV profiles) as dots. Data from each culture are shown in a different color. Horizontal bars represent the means of the averages from the three cultures, which were compared using a two-tailed unpaired *t*-test. Error bars are SD. Number of DCV profiles analyzed: 135 (control) and 64 (DKO). *p<0.05. (**C**) Frequency distribution of DCVs by diameter is skewed to the left in DKO neurons, indicating a higher proportion of smaller DCVs in DKO neurons than in control. (**D**) Example EM images show normal ultrastructure of synapses and other cellular organelles in DKO neurons. Golgi cisternae and selected mitochondria are labeled with 'G' and 'M,' respectively. Scale bar 100 nm. (**E–H**) Quantification of SV diameter, SV number per synapse, length of the active zone, and length of postsynaptic density (PSD) from EM images exemplified in (**D**). Data are shown as SuperPlots and analyzed as described in (**B**). Number of individual observations: 617–621 SV profiles (**E**) and 148–152 synapses (**F–H**). ns: not significant.

fusion protein into DCVs during the RUSH assay (*Figure 5—figure supplement 2*). Formation of such puncta and clearance of NPY-EGFP-SBP from the Golgi was observed in neurons of both genotypes. We next compared the speed of NPY-EGFP-SBP flux through the Golgi by plotting EGFP intensity within the Golgi area over time (*Figure 5F*). The peak of fluorescence intensity in the Golgi was reached 5.6 min faster in DKO than in control, indicating that the speed of Golgi filling is increased

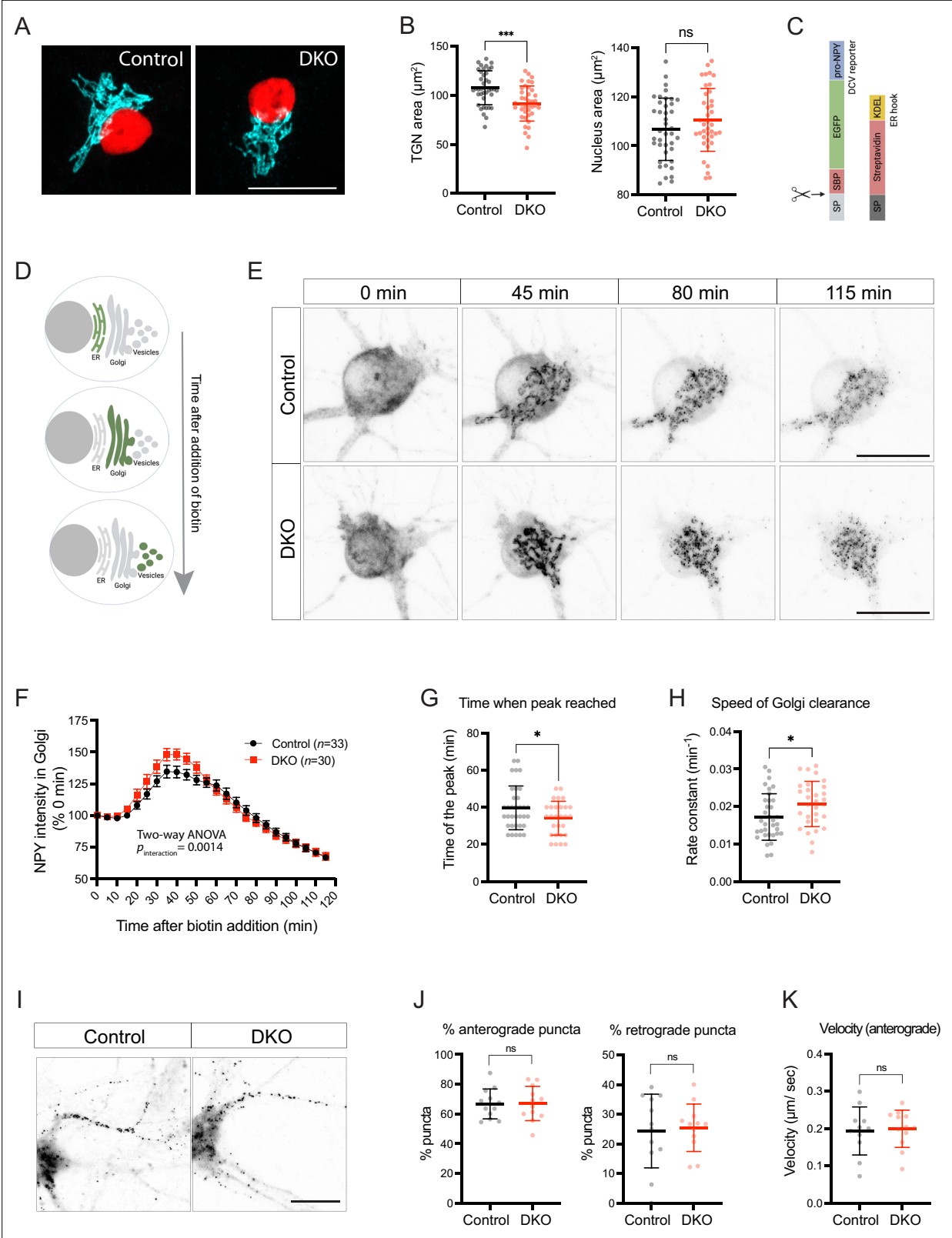

**Figure 5.** Loss of tomosyns results in the accelerated trafficking of a dense core vesicle (DCV) cargo through Golgi. (**A**) Trans-Golgi network (TGN) area is decreased in double knockout (DKO) neurons. Representative images of immunostained TGN marker, TMEM87A (shown in cyan). Nuclei are labeled in red due to the expression of mCherry-tagged Cre or ΔCre. Scale bar 20 μm. (**B**) Quantification of the area of the TGN and nuclei from confocal microscopy images exemplified in (**A**). Data are shown as mean ± SD and were analyzed using a two-tailed unpaired *t*-test. n=39–40 neurons/ genotype.

*Figure 5 continued on next page*

*Figure 5 continued*

***p<0.001; ns: not significant. (**C**) Design of the construct for the Retention Using Selective Hooks (RUSH) assay. As a DCV cargo, human pre-pro-NPY fused to streptavidin-binding peptide (SBP) and enhanced green fluorescent protein (EGFP) was used. The cargo was immobilized at the endoplasmic reticulum (ER) due to interaction with the ER hook, consisting of streptavidin fused to the ER-retention/retrieval signal KDEL. (**D**) Scheme illustrating synchronized trafficking of a DCV cargo through the secretory pathway upon the addition of biotin in the RUSH assay. (**E**) Snapshots of time-lapse videos taken after the addition of biotin in the RUSH assay. Time passed after the addition of biotin is shown on top. DCV cargo passes through the Golgi in roughly 45 min after the addition of biotin and gets concentrated in fine puncta that eventually leave cell soma. Scale bar 20 μm. (**F**) NPY-EGFP intensity in the Golgi area plotted against time after the addition of biotin. Intensity values were normalized to the basal levels (before the addition of biotin). Data are shown as mean ± SEM and were analyzed by two-way ANOVA. n=30–34 neurons/genotype. (**G**) Time required to reach maximum NPY-EGFP intensity at the Golgi, quantified from (**F**), is reduced in DKO. Data are shown as mean ± SD and were analyzed by a two-tailed unpaired *t*-test. *p<0.05. (**H**) Speed of NPY-EGFP export from the Golgi is increased in DKO neurons. Rate constants were calculated from first-order decay curves fitted to the data shown in (**F**). Data are shown as mean ± SD and were analyzed by a two-tailed unpaired *t*-test. *p<0.05. (**I**) Snapshots of time-lapse videos taken 90 min after the addition of biotin in the RUSH assay with the focus on neurites, where newly made DCVs trafficked after exit from the Golgi. (**J**) Proportion of newly made DCVs trafficking in anterograde and retrograde directions as quantified from the time-lapse videos exemplified in (**I**). Majority of vesicles are trafficked anterogradely in neurons of both genotypes. Data are shown as mean ± SD and were analyzed by a two-tailed unpaired *t*-test. n=12 neurons/genotype. ns: not significant. (**K**) Speed of anterogradely trafficking vesicles as quantified from the time-lapse videos exemplified in (**I**). Data are shown as mean ± SD and were analyzed by a two-tailed unpaired *t*-test. n=12 neurons/genotype. ns: not significant.

The online version of this article includes the following video and figure supplement(s) for figure 5:

**Figure supplement 1.** Localization of tomosyn to the trans-Golgi network (TGN), dense core vesicles (DCVs), and synapses.

**Figure supplement 2.** Retention Using Selective Hooks (RUSH)-generated vesicles contain endogenous dense core vesicle (DCV) marker, chromogranins (CHGA).

**Figure 5—video 1.** Live imaging of dense core vesicle (DCV) formation in the Retention Using Selective Hooks (RUSH) assay.

https://elifesciences.org/articles/85561/figures#fig5video1

in DKO (*Figure 5G*). The speed of Golgi clearance – determined by fitting a first order exponential decay curve to the data after the peak – was also increased (by 20%) in DKO neurons (*Figure 5H*). Thus, loss of tomosyns does not impede export of the overexpressed neuropeptide cargo from Golgi but, conversely, accelerates it.

We also traced the fate of newly made RUSH vesicles 90 min after the addition of biotin. After the exit from Golgi, these vesicles traveled predominantly in one neurite, morphologically identified as the axon (*Figure 5I*). Most axonal vesicles (67%) traveled in the anterograde direction with a similar speed in neurons of both genotypes (*Figure 5J–K*), indicating that the transport of newly made DCVs is not affected by the loss of tomosyns.

We then asked if the loss of tomosyns affected mRNA expression of secretory cargos that were significantly decreased in DKO as evidenced by mass spectrometry and WB analysis. We detected a decrease in the expression of various transcript variants of *Bdnf* (*Figure 6A*) and other common DCV cargos, most prominently *Vgf* and *Pcsk1*, encoding a prohormone convertase (*Figure 6B*). Interestingly, the decrease in mRNA was less pronounced than the decrease at the protein level, as detected by WB (for BDNF) or mass spectrometry (for VGF and PCSK1) (*Figure 6—figure supplement 1*). The number of neuropeptides was decreased at the mRNA level in DKO as well (*Figure 6C*). *Grp* (gastrin related peptide) and *Pnoc* (nociceptin, nocistatin) showed the strongest reduction (by 50% and 35%, respectively), while mRNA of genes encoding cholecystokinin, NPY, and tachykinins displayed a reduction by around 30%. It is not clear whether any of these peptides were also decreased at the protein level, since most processed neuropeptides could not be detected in our mass spectrometry-based proteomics analysis due to their small size. Based on single-cell transcriptomics data (*Saunders et al., 2018*), expression of the affected genes was not confined to a specific neuronal population (e.g. glutamatergic or GABAergic neurons). Strikingly, re-expression of tomosyn (isoform 1 m) from DIV2 onwards did not restore altered transcription of the affected genes, suggesting that tomosyn does not mediate these changes (*Figure 6D*). Since no annotated genes and functional elements, other than *Stxbp5* and *Stxbp5l*, were predicted to be affected by recombination of the *Stxbp5*/*Stxbp5*l loci in our mouse model, we tested whether the expression of Cre recombinase per se results in the detected transcriptional defects. Indeed, Cre expression in wild-type neurons not bearing introduced loxP sites led to a decrease in the mRNA expression of *Bdnf*, *Vgf*, *Pcsk1*, and *Grp* when compared to ΔCre-expressing wild-type neurons, with the size effect comparable to the one observed in *Stxbp5/5l*^lox neurons (*Figure 6E*). Expression of *Gapdh*, used as housekeeping gene control, was not affected by

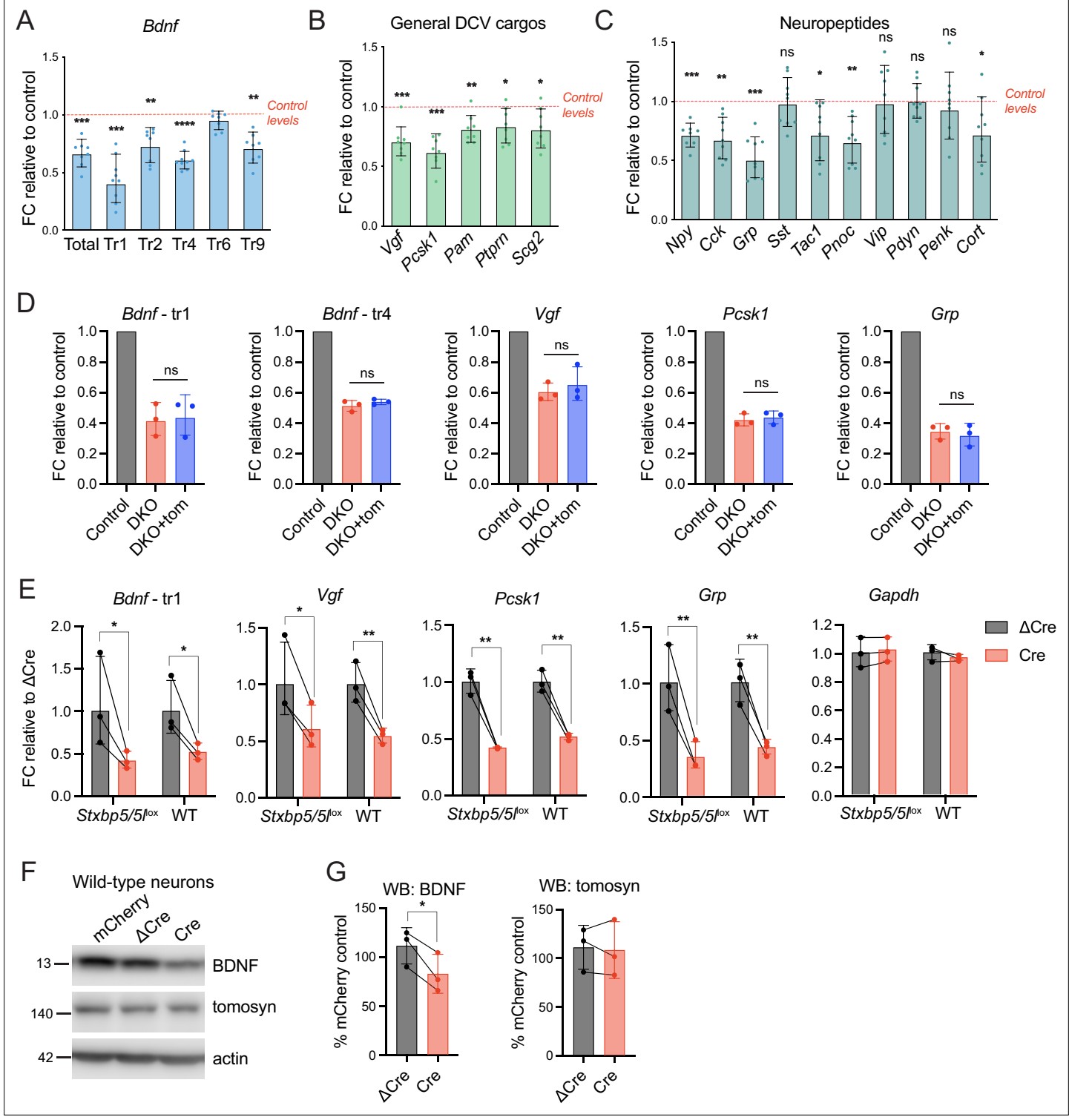

**Figure 6.** Decreased mRNA expression of some dense core vesicle (DCV) cargos in double knockout (DKO) is caused by Cre recombinase expression and not by the loss of tomosyns. (**A**) mRNA levels of different *Bdnf* transcript isoforms as assessed by quantitative RT-PCR and normalized to *Gapdh*. Data are shown as fold changes (FC) relative to control levels in the corresponding culture preparation. n=8 culture preparations/genotype. Dots represent individual cultures, bar graphs are geometric means, and error bars are geometric SD. Log2FC (ΔΔCt) were analyzed by one-sample *t*-test. **p<0.01; ***p<0.001; ****p<0.0001. (**B**) mRNA levels of general DCV cargos as assessed by quantitative RT-PCR and normalized to *Gapdh*. Data are shown and analyzed as described in (**A**). *p<0.05; **p<0.01; ***p<0.001. (**C**) mRNA levels of most abundant hippocampal neuropeptides as assessed by quantitative RT-PCR and normalized to *Gapdh*. Neuropeptides are sorted by their abundance in hippocampal neurons. Data are shown and analyzed

*Figure 6 continued on next page*

*Figure 6 continued*

as described in (**A**). *p<0.05; **p<0.01; ***p<0.001, ns: not significant. (**D**) Re-expression of tomosyn does not restore mRNA expression of DCV cargos in DKO neurons. mRNA levels of *Bdnf*, *Vgf*, *Pcsk1*, and *Grp* were normalized to *Gapdh* and plotted as described in (**A**). Comparison between DKO and DKO re-expressing tomosyn (DKO +tom) is performed using a two-tailed paired *t*-test on Log2FC (ΔΔCt). n=3 culture preparations/ genotype. ns: not significant. (**E**) Expression of Cre-recombinase in wild-type (WT) neurons results in the decreased mRNA expression of DCV cargos. Effects of Cre on *Stxbp5/5l*^lox^ neurons are shown for the direct comparison. mRNA levels of the DCV cargos were normalized to *Gapdh* and plotted as fold changes (FC) relative to ΔCre. Comparisons between ΔCre and Cre conditions in each genotype were performed using a two-tailed paired *t*-test. n=8 culture preparations/genotype. *p<0.05; **p<0.01. (**F**) brain-derived neurotrophic factor (BDNF) and tomosyn levels as detected by western blot (WB) in lysates of ΔCre-mCherry and Cre-mCherry-expressing wild-type neurons. mCherry expression was used as a control. Equal loading was verified by immunodetection of GAPDH. (**G**) Quantification of BDNF and tomosyn bands intensity from WB exemplified in (**F**). BDNF and tomosyn levels in (Δ) Cre-expressing wild-type neurons are shown as mean % of control (mCherry-expressing neurons). Error bars represent SD. Data were analyzed using a two-tailed paired *t*-test. n=3 samples/ genotype. *p<0.05.

The online version of this article includes the following source data and figure supplement(s) for figure 6:

**Source data 1.** Uncropped western blot (WB) images for *Figure 6F*.

**Figure supplement 1.** Comparison of mRNA and protein levels of brain-derived neurotrophic factor (BDNF), VGF, and PCSK1 in double knockout (DKO) neurons.

**Figure supplement 2.** Cre recombinase does not affect levels of the overexpressed dense core vesicle (DCV) reporter (NPY-pHluorin) in wild-type neurons.

Cre. Thus, changes in DCV cargos transcription detected in DKO neurons are caused by direct effects of Cre expression, rather than by loss of tomosyns.

We next asked whether this effect of Cre expression contributes to the decrease of DCV cargos in DKO at the protein level. Lentivirus-mediated expression of Cre in wild-type neurons resulted in a decrease of BDNF levels by 26% (*Figure 6F–G*), as compared to >50% reduction in BDNF documented in DKO (Cre-expressing *Stxbp5/5l*^lox^) neurons (*Figure 2E–F*). Expression of ΔCre that lacks the catalytic domain did not affect BDNF levels when compared to wild-type neurons expressing mCherry under the same promoter (*Figure 6F–G*), suggesting that recombinase activity is necessary for the effect of Cre on BDNF levels. Cre expression in wild-type neurons did not affect tomosyn levels. These data indicate that unspecific Cre recombinase activity contributes to the reduction in BDNF protein levels in DKO neurons, but to a lesser extent than the loss of tomosyns. In line with this conclusion, BDNF protein levels in DKO were largely rescued by the re-expression of tomosyn (*Figure 2C–D and G–H*).

To test if the latter conclusion also applies to a DCV reporter overexpressed under the control of a synapsin promoter, we tested the effects of Cre on the levels of NPY-pHluorin. Cre expression in wild-type neurons did not affect levels of NPY-pHluorin as shown by ICC (*Figure 6—figure supplement 1*), while the same reporter showed an approximately 20% reduction upon expression of Cre in *Stxbp5/5l*^lox^ neurons that was rescued by re-expression of tomosyn (*Figure 2A–B*). Thus, while Cre expression impairs transcription of endogenous DCV cargos in both wild-type and *Stxbp5/5l*^lox^ neurons, it affects protein levels of the overexpressed DCV reporter only in *Stxbp5/5l*^lox^ but not in wild-type neurons.

## Discussion

In this study, we examined the role of tomosyns in the exocytosis of neuropeptide-containing DCVs in mammalian neurons. Using a pHluorin-based DCV fusion reporter, we found that the conditional loss of tomosyns does not alter the number of DCV exocytosis events but lowers intracellular levels of the DCV reporter. We corroborated the latter finding by showing that levels of endogenous DCV cargos, such as BDNF, are decreased in tomosyn DKO neurons by three independent approaches (ICC, WB, and mass spectrometry). The effect on the DCV pool and/or composition was accompanied by a smaller DCV size and increased speed of the DCV reporter flux through the TGN. Taken together, these data suggest that tomosyn is dispensable for control of DCV fusion, but, paradoxically, has an impact on the intracellular levels of DCV cargos.

The lack of the effect on neuronal DCV fusion is surprising, given that tomosyn regulates SV release in neurons and secretory granule exocytosis in neuroendocrine cells and platelets. However, phenotypes caused by the loss of tomosyn expression in different cell types are discrepant. While the loss

of tomosyn resulted in enhanced SV fusion in the mouse brain and invertebrate nervous system (*Sakisaka et al., 2008*; *Ben-Simon et al., 2015*; *Gracheva et al., 2006*; *McEwen et al., 2006*; *Chen et al., 2011*; *Sauvola et al., 2021*), tomosyn deficiency in platelets, INS-1E cells and rat superior cervical ganglion neurons led to an opposite phenotype – a strong reduction in secretion (*Ye et al., 2014*; *Zhu et al., 2014*; *Cheviet et al., 2006*; *Baba et al., 2005*). No unifying model explains the discrepant effects of tomosyn on secretory vesicle fusion. Altogether, the discrepant effects of tomosyn on fusion in different cell types suggest that tomosyn does not act solely as an inhibitor of fusogenic SNARE complex formation.

DCVs and SVs differ in release properties, and recent studies shed light on some components of the molecular machinery contributing to these differences (*Persoon et al., 2019*; *Moro et al., 2021*). In contrast to SVs, which are released in response to a single electric pulse, neuronal DCVs require prolonged high-frequency stimulation for induction of exocytosis (*Hartmann et al., 2001*; *Xia et al., 2009*; *Persoon et al., 2018*). Several factors can explain the reluctance of DCVs to fuse, among them the loose coupling of DCVs to the sites of calcium entry, requiring an increase in calcium levels reaching beyond an immediate radius of a calcium channel (*Mansvelder and Kits, 2000*). The difference in calcium sensitivity between SVs and DCVs may facilitate frequency-dependent release of different types of chemical signals from neurons and ensure that neuropeptides are secreted predominantly at high firing rates. Interestingly, the inhibitory effect of tomosyn on DCV fusion in adrenal chromaffin cells is calcium-dependent and could be overridden by an increase in intracellular calcium (*Yizhar et al., 2004*). Thus, it is possible that high-frequency stimulation and a massive increase in intracellular calcium required for the induction of neuronal DCV exocytosis overrides the regulatory effect of tomosyn.

Despite the lack of effect on DCV exocytosis, loss of tomosyns resulted in the decreased intracellular levels of overexpressed (NPY-pHluorin) and endogenous (BDNF, IA-2) DCV cargos (*Figure 2*, *Figure 2—figure supplement 1*). Importantly, this phenotype was reproduced in DKO neurons grown under three different culture conditions that vary in cell density and the ratio of neurons to glia (single neurons grown on glial microislands, sparse network cultures on glial feeder layer, and dense network cultures on coated plastic). This phenotype is consistent with the effect of tomosyn's loss in platelets, i.e. a strong decrease in many secretory cargos, despite the fact that the release of these cargos was reduced, not enhanced (*Ye et al., 2014*). Tomosyn-deficient platelets showed more than 50% decrease in the levels of alpha-granules cargos, such as factor V and platelet factor 4 (PF4), although alpha-granule count was not affected. Similarly, loss of tomosyn orthologs SRO7/77 in yeast resulted in the altered composition of secretory vesicles and re-routing of a secretory cargo, sodium pumping ATPase Ena1p, from secretory vesicles into the degradative pathway (vacuole) (*Forsmark et al., 2011*; *Wadskog et al., 2006*). Despite the fact that alpha-granules in platelets and secretory vesicles in yeast differ from neuronal DCVs in many aspects, taken together, our published data suggest a general role of tomosyn and its orthologs in sorting cargos into various types of secretory granules. The enrichment of tomosyn at the TGN (*Figure 5—figure supplement 1*) and decreased DCV and Golgi size in DKO neurons (*Figures 4A–C and 5A–B*) are consistent with such a role.

Importantly, the loss of tomosyns in neurons did not abolish the formation of new secretory granules, as evidenced by the normal budding of NPY-vesicles in the RUSH assay (*Figure 5E*, *Figure 5—video 1*). However, NPY flux through the Golgi was slightly accelerated in DKO (*Figure 5F–H*), supporting the hypothesis that tomosyn regulates the sorting of the soluble cargo at the TGN. The decreased size of DCVs suggests that loading of DCVs with secretory cargo is impaired in DKO, since DCV size is regulated by the amount of loaded cargo, at least in non-neuronal cell types (*Clermont et al., 1993*; *Boquist and Lorentzon, 1979*). On the other hand, the peak amplitude of individual NPY-pHluorin fusion events, an estimate of NPY loading per fusing DCV, was not affected in DKO (*Figure 1G*). This discrepancy suggests that smaller DCVs in DKO may be reluctant to fuse due to the deficiency in specific transmembrane cargos necessary for DCV exocytosis. Another important observation is that interfering with lysosomal activity did not rescue cargo levels in DKO neurons (*Figure 2G*), indicating that cargo is not mis-sorted to lysosomes in the absence of tomosyns.

Re-expression of tomosyn rescued protein, but not mRNA expression, of DCV cargos in DKO neurons (*Figure 6D*). We found that the recombinase activity of Cre, irrespective of the presence of the loxP sites, affects the expression of some DCV cargos (*Bdnf*, *Grp*, *Pcsk1*), explaining the inability of tomosyn to rescue transcription of these genes. It has been long acknowledged that Cre expression

can have deleterious side effects (*Schmidt et al., 2000*; *Loonstra et al., 2001*; *Heidmann and Lehner, 2001*; *Forni et al., 2006*; *Naiche and Papaioannou, 2007*), although, to our knowledge, none were reported for primary neuronal cultures. Cre can catalyze recombination between pseudo (cryptic) lox*P* sites (*Thyagarajan et al., 2000*). Our data suggest that Cre expression in postmitotic neurons affects neuronal transcriptome and warrants caution when interpreting data on the proteome DKO neurons.

Notwithstanding this effect, the current study provides robust evidence that the decrease of BDNF at the protein level is caused, to a large extent, specifically by the loss of tomosyns, and not by the effects of Cre on transcription. First, BDNF levels were decreased by more than 50% in DKO neurons (*Figure 2C–D and E–F*), and only by around 26% in Cre-expressing wild-type neurons (*Figure 6F–G*). Second, BDNF levels in DKO were restored to approximately 80% of control levels by re-expression of tomosyn (*Figure 2C–D*). Finally, in support of the specific effects of tomosyn on DCV cargos, levels of the overexpressed NPY-pHluorin levels were decreased in DKO, but not in Cre-expressing wild-type neurons, and this decrease was rescued by the re-expression of tomosyn (*Figure 2A–B*, *Figure 6— figure supplement 1*).

Interestingly, the SNARE domain of tomosyn was dispensable for the rescue of BDNF levels in DKO, suggesting that the N-terminal WD40-domain mediates this effect of tomosyn (*Figure 2G–H*). The main structural feature of the WD40-domain are two bulky beta-propellers comprised of multiple WD40 repeats. This domain constitutes the largest part of tomosyn's molecule, and it is best conserved among all tomosyn orthologs, including the yeast SRO7/77 that, in fact, lack the canonical SNARE domain. Previous studies showed that WD40-domain alone is not sufficient to rescue synaptic transmission defects caused by the loss of tomosyn orthologs in the fly and nematode (*Burdina et al., 2011*; *Sauvola et al., 2021*). On the other hand, overexpression of the WD40 domain alone has an inhibitory effect on secretion from bovine chromaffin and rat pheochromocytoma PC12 cells (*Yizhar et al., 2007*; *Bielopolski et al., 2014*). This effect has been attributed to the binding of specific loops in the WD40 propeller region to SNAP25 (*Bielopolski et al., 2014*). Several other interaction partners of tomosyn were reported to bind to the WD40-propellers, such as Rab3A and synaptotagmin (*Yamamoto et al., 2010*; *Cazares et al., 2016*), suggesting that the WD40 domain may serve as a scaffold for the assembly of the dynamic protein complexes. The strength of these interactions can be regulated by posttranslational modifications of the WD40 domain, such as phosphorylation (by PKA, ROCK) and SUMOylation (*Baba et al., 2005*; *Gladycheva et al., 2007*; *Ferdaoussi et al., 2017*). A recent study identified tomosyn as an inhibitor of RhoA in the WD40-domain dependent manner (*Shen et al., 2020*). RhoA is a small GTPase that regulates, among other processes, membrane trafficking at the Golgi (*Quassollo et al., 2015*). Remarkably, Sro7 was originally identified in a genome-wide screen as a multicopy suppressor of RHO3, an ortholog of Rho GTPases in yeast, suggesting an evolutionary conserved link between tomosyn and Rho GTPases (*Matsui and Toh-E, 1992*). Further studies are required to validate the potential regulation of Rho by tomosyn and to assess a role of this hypothetical pathway at the Golgi.

## Materials and methods

**Key resources table**

| Reagent type (species) or resource | Designation | Source or reference | Identifiers | Additional information |
|---|---|---|---|---|
| Gene (*Mus musculus*) | Stxbp5 | NCBI | 78808 | syntaxin binding protein 5 (tomosyn) |
| Gene (*Mus musculus*) | Stxbp5l | NCBI | 207227 | syntaxin binding protein 5-like (tomosyn-2) |
| Genetic reagent (*Mus musculus*) | C57BL/6J-*Stxbp5*^lox^/*Stxbp5l*^lox^ | This paper | - | Generation of the mouse model is described in Materials and Methods (Mice) |
| Genetic reagent (*Mus musculus*) | C57BL/6 J | Charles River | Strain code 631 | |
| Genetic reagent (*Rattus norvegicus*) | Wistar (Crl:WI) | Charles River | Strain code: 003 | Used for preparation of glia feeder layer |
| Antibody | anti-syntaxin (mouse monoclonal) | Sigma | S0664 | 1:2000 |

*Continued on next page*

*Continued*

| Reagent type (species) or resource | Designation | Source or reference | Identifiers | Additional information |
|---|---|---|---|---|
| Antibody | anti-SNAP25 (mouse monoclonal) | Covance | SMI-81R | 1:1000 |
| Antibody | anti-VAMP2 (mouse monoclonal) | Synaptic Systems | 104211 | 1:2000 |
| Antibody | anti-tomosyn (rabbit polyclonal) | Synaptic Systems | 183103 | 1:1000 (WB) 1:500 (ICC) |
| Antibody | anti-tomosyn-2 (rabbit polyclonal) | Synaptic Systems | 183203 | 1:1000 |
| Antibody | anti-alpha-tubulin (mouse monoclonal) | Synaptic Systems | 302211 | 1:1000 |
| Antibody | anti-actin (mouse monoclonal) | Merck | MAB1501 | 1:1000 |
| Antibody | anti-synaptophysin (guinea pig polyclonal) | Synaptic Systems | 101004 | 1:1000 |
| Antibody | anti-MAP2 (chicken polyclonal) | Abcam | ab5392 | 1:5000 |
| Antibody | anti-IA-2 (PTPRN) (mouse monoclonal) | Merck | MABS469 | 1:100 |
| Antibody | anti-CHGA (rabbit polyclonal) | Synaptic Systems | 259003 | 1:500 |
| Antibody | anti-CHGB (rabbit polyclonal) | Synaptic Systems | 259103 | 1:500 |
| Antibody | anti-BDNF (mouse monoclonal) | DSHB hybridoma product | BDNF #9 (supernatant)[1] | 1:4 |
| Antibody | anti-BDNF (mouse monoclonal) | Biosensis | BSENM-1736–50 (clone 3C11) | 1:500 |
| Antibody | anti-LAMP1 (rat monoclonal) | DSHB hybridoma product | 1D4B (supernatant)[2] | 1:20 (ICC) 1:200 (WB) |
| Antibody | anti-LIMP2 (rabbit polyclonal) | Novus Biologicals | NB400-129 | 1:1000 |
| Antibody | anti-GAPDH (rabbit polyclonal) | Elabscience | E-AB-40337 | 1:2000 |
| Antibody | anti-TMEM87A (rabbit polyclonal) | Novus Biologicals | NBP1-90532 | 1:100 |
| Antibody | anti-TGN38 (sheep polyclonal) | Bio-Rad | AHP499G | 1:100 |
| Recombinant DNA reagent | pLenti-Syn(pr)-Cre-mCherry | *Wolzak et al., 2022* | - | |
| Recombinant DNA reagent | pLenti-Syn(pr)-ΔCre-mCherry | *Wolzak et al., 2022* | - | Cre lacking catalytic domain |
| Recombinant DNA reagent | pLenti-Syn(pr)-pre-NPY-pHluorin | *Nassal et al., 2022* | - | |
| Recombinant DNA reagent | pLenti-mScarlet-T2A-tomosyn | This paper | - | mouse isoform m (NM_001081344). Generation of this reagent is described in Materials and methods (Expression constructs) |
| Recombinant DNA reagent | pLenti-mScarlet-T2A-tomosyn(aa1-1047) | This paper | - | Generation of this reagent is described in Materials and methods (Expression constructs) |
| Recombinant DNA reagent | pLenti-Syn(pr)-mCherry | This paper | - | Generation of this reagent is described in Materials and methods (Expression constructs) |
| Recombinant DNA reagent | pCMV-Streptavidin-KDEL-IRES-EGFP-NPY | *Emperador-Melero et al., 2018* | - | Construct used for RUSH assay |
| Sequence-based reagent | qPCR primers are listed in *Table 1* | | | |

*Continued on next page*

*Continued*

| Reagent type (species) or resource | Designation | Source or reference | Identifiers | Additional information |
|---|---|---|---|---|
| Peptide, recombinant protein | 2.5% trypsin | gibco | 15090046 | |
| Peptide, recombinant protein | papain | Worthington Biochemical Corporation | LS003127 | |
| Peptide, recombinant protein | poly-L-ornithine | Sigma | P4957 | |
| Peptide, recombinant protein | laminin | Sigma | L2020 | |
| Peptide, recombinant protein | rat tail collagen | BD Biosciences | 354236 | |
| Peptide, recombinant protein | poly-D-lysine | Sigma | P6407 | |
| Commercial assay or kit | SensiFast cDNA Synthesis Kit | Meridian Bioscience | BIO-65054 | |
| Commercial assay or kit | SensiFast SYBR Lo-ROX Kit | Meridian Bioscience | BIO-94020 | |
| Chemical compound, drug | leupeptin | Hello Bio | HB3958 | |
| Chemical compound, drug | biotin | Sigma | B4501 | |
| Software, algorithm | MATLAB | MathWorks | - | |
| Software, algorithm | Prism | GraphPad | - | |
| Software, algorithm | Fiji/ImageJ | NIH | - | |

## Mice

*Stxbp5*$^{lox/lox}$, *Stxbp5l*$^{lox/lox}$ double conditional null mice were generated by breeding two lines with the individually targeted *Stxbp5* and *Stxbp5l* alleles. To generate *Stxbp5*$^{lox/+}$ mice, *lox2272* recombination sites flanking exon 2 of *Stxbp5* (98 bp in size) were introduced into the genome via homologous recombination in C57Bl/6 embryonic stem (ES) cells (Cyagen Animal Model Services, Santa Clara, USA). Lox2272 recombines efficiently with other lox2272 sites, but not with loxP (*Lee and Saito, 1998*). Deletion of exon 2 causes a frameshift in all tomosyn reference transcripts (NM_001081344.3, NM_001408063.1, NM_001408064.1, and NM_001408065.1). The targeting vector contained a neomycin resistance marker, flanked by self-deleting anchor sites and placed between exon 2 and the lox2272 site in intron 2. Targeted ES clones were injected into blastocysts to produce germ-line chimeras, which were mated to C57Bl/6 J wild-type mice. *Stxbp5* line genotyping was performed using the following primers (5′ ACTTAGCGCGGAGGGTTTTGTC 3′ and 5′ CGTAGGCTTTTAAATCACCG CTGT 3′) to amplify *Stxbp5*$^+$ or S*txbp5*$^{lox}$ specific products of 196 and 236 bp in size, respectively.

The *Stxbp5l*$^{lox/+}$ line was generated independently as described in *Geerts et al., 2015*. In these mice, the paralogous exon 3 (also 98 bp in size) is flanked by loxP sites. Cre-mediated excision induces a frameshift in all reference transcripts (NM_172440.3, NM_001114611.1, NM_001114612.1, NM_001114613.1, also named xb-, s-, b- and m-tomosyn-2, respectively). *Stxbp5l* line genotyping was performed for *Stxbp5l* as described (*Geerts et al., 2015*).

After interbreeding the two mouse lines, both backcrossed to an inbred C57Bl/6 J genetic background, newborn pups from homozygous *Stxbp5*$^{lox/lox}$, *Stxbp5l*$^{lox/lox}$ matings were used for the preparation of neuronal cultures in all the described experiments. Mice were housed and used for experiments according to institutional guidelines and Dutch laws in accordance with the European Communities Council Directive (2010/63/EU).

## Neuronal cultures

Mouse hippocampal cultures were prepared from newborn pups. Normally, hippocampi from several pups from one nest were pooled in one culture preparation, which was considered as one biological

replicate. Hippocampi were dissected in Hanks Balanced Salt Solution (HBSS, Sigma H9394) supplemented with 10 mM HEPES (pH 7.2–7.5, Gibco 15630056) and digested with 0.25% trypsin (Gibco 15090046) at 37 °C for 15 min. After the digestion, tissue pieces were washed in HBSS and triturated using fire-polished Pasteur pipettes in Dulbecco's modified Eagle medium (DMEM, VWR Life Science VWRC392-0415) supplemented with 10% fetal calf serum (FCS, Gibco 10270), non-essential amino acids (Sigma M7145), and antibiotics (penicillin/streptomycin, Gibco 15140122). For most immuno-cytochemistry (ICC) experiments, cell suspension was sparsely plated on a feeder layer of rat glia at a density of $20–30 \times 10^3$/well on 15 mm coverslips. For BDNF ICC and pHluorin imaging experiments, neurons were plated on glial micro-islands at a density of 1500/well (to obtain single neuron cultures). For RUSH, rat glia was omitted, and neurons were plated on glass coverslips at a density of approximately $120 \times 10^3$/well on 15 mm coverslips. Prior to plating, glass coverslips were coated with poly-L-ornithine (Sigma P4957) and laminin (Sigma L2020). For western blot, proteomics, electron microscopy, and qPCR experiments, neurons were plated on coated plastic without the glia feeder layer at a density of $400 \times 10^3$/well in a six-well plate. Cultures were maintained in a humified incubator at 37 °C and 5% $CO_2$ in Neurobasal medium (Gibco 21103049) supplemented with B27 (Gibco 17504044), 10 mM HEPES, GlutaMAX (Gibco 35050038) and antibiotics (penicillin/streptomycin, Gibco 15140122). To induce loss of tomosyns, cultures were transduced within 8 hr after plating with a lentivirus encoding Cre-recombinase expressed under the control of synapsin promoter (pSyn-Cre-mCherry). As a control, neurons from the same culture preparation were transduced with *Cre*-recombinase lacking the catalytic domain (pSyn-ΔCre-mCherry). Experiments were performed, if not otherwise stated, on day in vitro (DIV) 14. For rescue experiments, neurons were transduced on DIV2 with lentivirus encoding either FL tomosyn (isoform 1 m, NM_001081344, aa 1–1116) or truncated tomosyn (aa1-1047) lacking the SNARE domain.

Rat glia were prepared from the cortices of newborn rats. Cortices were digested in papain (Worthington Biochemical Corporation LS003127) at 37 °C for 45 min and triturated in DMEM supplemented with 10% FCS, non-essential amino acids, and antibiotics. Glia were plated and expanded in T175 flasks. For making single neuron cultures, glia were plated on micro-islands of growth permissive substrate (mix of collagen I and poly-D-lysine, Corning 354236 and Sigma P6407, respectively) that were printed on a layer of 0.15% agarose.

## Expression constructs

All transgenes, except for the RUSH reporter, were expressed under the control of the human synapsin promoter in lentiviral backbones. Cre-mCherry and ΔCre-mCherry fusions were described by *Wolzak et al., 2022*. Generation of the DCV fusion reporter (pre-NPY-pHluorin) was described by *Nassal et al., 2022*. Constructs encoding FL tomosyn (NM_001081344, aa1-1116) and truncated mutant (aa1-1047) of tomosyn lacking the SNARE domain were generated in this study. Tomosyn-encoding sequences were amplified from the mouse cDNA library and cloned with the N-terminal mScarlet succeeded by the T2A sequence into the lentiviral backbone. Construct encoding constitutively active CREB mutant (Y134F) was described in *Moro et al., 2020*. RUSH reporter (Streptavidin-KDEL-IRES-SBP-EGFP-NPY) was expressed under the control of the CMV promoter in pIRESneo3 backbone. Generation of this reporter was described in *Emperador-Melero et al., 2018*.

## Visualization and analysis of DCV fusion

Single neuron cultures were transduced on DIV8 with a lentivirus encoding the N-terminal part of human NPY fused to super-ecliptic pHluorin (*Figure 1—figure supplement 2*). DIV14-15 neurons were used for live imaging on a Nikon Ti Eclipse inverted microscope equipped with A1R confocal module and Andor DU-897 camera (with 512x512 pixels frame) under 40 x oil objective (numerical aperture, NA 1.3). LU4A laser unit with 488 nm wavelength laser was used as the illumination source. Coverslips were transferred in an imaging chamber in Tyrode's solution (119 mM NaCl, 2.5 mM KCl, 2 mM $CaCl_2$, 2 mM $MgCl_2$, 25 mM HEPES, and 30 mM Glucose, pH 7.4, 280 mOsmol) and time-lapse videos were recorded at 2 Hz frequency for 2 min at room temperature under constant perfusion with Tyrode's. After 30 s of baseline recording, neurons were subjected to electric field stimulation (16 trains, each consisting of 50 pulses at 50 Hz and separated by 500 msec intervals) delivered by platinum electrodes. At the end of each recording, 50 mM $NH_4Cl$-containing Tyrode's solution (with NaCl

**Table 1.** List of qRT-PCR primers used in this study.

| Gene | Forward primer sequence | Reverse primer sequence |
| --- | --- | --- |
| *Bdnf* (all transcripts) | GGCTGACACTTTTGAGCACGTC | CTCCAAAGGCACTTGACTGCTG |
| *Bdnf* (transcript 1) | ACTGAGCAAAGCCGAACTT | TCTCACCTGGTGGAACATTGTG |
| *Bdnf* (transcript 2) | TAGGCTGGAATAGACTCTTGG | CTCACCTGGTGGAACTTCTTTG |
| *Bdnf* (transcript 4) | CCTCCCCCTTTTAACTGAAG | CTCACCTGGTGGAACTTTTT |
| *Bdnf* (transcript 6) | AGGGACCAGAAGCGTGACAA | CTCACCTGGTGGAACTCAGGGT |
| *Bdnf* (transcript 9) | TGATTGTGTTTCTGGTGACA | CGGTTTCTAAGCAAGTGAAC |
| *Vgf* | CTTTGACACCCTTATCCAAGGCG | GCTAATCCTTGCTGAAGCAGGC |
| *Pcsk1* | GGAGAGAATCCTGTAGGCACCT | GCTCTGGTTGAGAAGATGTCCC |
| *Pam* | AGTCGGATCGTGCAGTTCTCAC | ACTGGTTCAGGTGAGGCACAAG |
| *Ptprn* | TGGCAGGCTATGGAGTAGAGCT | CTTGACATCGGCTCCTCCAACA |
| *Scg2* | CAGGAAGAGGTGAGAGACAGCA | TGGAGGCATCCTCTGAGAGTTG |
| *Npy* | TACTCCGCTCTGCGACACTACA | GGCGTTTTCTGTGCTTTCCTTCA |
| *Cck* | GAGGTGGAATGAGGAAACAA | CAGATTTCACATTGGGGACT |
| *Grp* | TTCAAACCGCTAAGTTGGT | GAAGGGTTTTGTTTTGCTCC |
| *Sst* | TCTGGAAGACATTCACATCC | TTCTAATGCAGGGTCAAGTT |
| *Tac1* | AGATCTCTCACAAAAGGCAT | CATCGCGCTTCTTTCATAAG |
| *Pnoc* | TCCTCTTTTGTGACGTTCTG | GAGGATGCACGTCTTTAAGT |
| *Vip* | ATGATGTGTCAAGAAATGCC | ATTCGTTTGCCAATGAGTGA |
| *Pdyn* | ATCAACCCCCTGATTTGCTC | ATCTTCCAAGTCATCCTTGC |
| *Penk* | ACATCAATTTCCTGGCGTGC | TGTTATCCCAAGGGAACTCG |
| *Cort* | CTTCTTATGTCAGCTTTGCC | CTCCAATCCCTTAGTTGACC |
| *Gapdh* | CATCACTGCCACCCAGAAGACTG | ATGCCAGTGAGCTTCCCGTTCAG |

concentration reduced to 69 mM to maintain osmolarity) was applied to visualize the intracellular pool of DCVs.

Analysis of the time-lapse recordings was performed as described (*Persoon et al., 2019*). In short, fusion events were detected by a sudden rise in fluorescence intensity. 3 × 3 pixel regions of interest (ROI) were placed on time-lapse recordings using a custom-made script in Fiji (*Schindelin et al., 2012*). Change in fluorescence intensity (F) within these ROIs was plotted as $\Delta F/F_0$, where $F_0$ is an average of the first 10 frames of acquisition. Resulting traces were checked using a custom-made script in MATLAB, and only events with $F/F_0 \geq 2$ SD and rise time <1 s were counted. Total number of intracellular vesicles (intracellular pool) was determined as the number of fluorescent puncta after the ammonium pulse corrected to account for overlapping puncta. Released fraction was calculated as the number of fusion events per neuron divided by the intracellular pool of DCVs.

## Immunocytochemistry

Neurons grown on coverslips were fixed by the application of phosphate-buffered saline (PBS) containing 4% paraformaldehyde (PFA, Sigma P6148) and 4% sucrose for 10 min. After fixation, neurons were washed and the residual PFA was quenched by application of 0.1 M glycine for 5 min. Neurons were permeabilized and blocked in PBS containing 0.1% Triton X-100 (Fisher Chemical T/3751/08) and 2% normal goat serum (Gibco 16210–072) ('blocking buffer') for 30 min. Primary antibodies diluted in blocking buffer were applied for 1 hr at room temperature. Primary antibodies used in this study and their dilutions are listed in *Table 1*. After washing with PBS, neurons were incubated with Alexa Fluor-coupled secondary antibodies diluted at 1:500 for 45 min. Stained coverslips were mounted in Mowiol.

## Confocal microscopy of fixed neurons

Imaging of stained neurons was performed on a Nikon Ti Eclipse inverted microscope equipped with A1R confocal module and LU4A laser unit. Images were acquired under 40 x (NA 1.3) or 60 x (NA 1.4) oil objectives in the confocal mode. Acquisition parameters (zoom, image size, scanning speed, laser power, and gain) were set using NIS software to achieve appropriate image resolution and avoid signal saturation. Three to five z-sections with an interval of 0.2 μm were collected per image.

Analysis of staining intensity was performed on maximum intensity projection of z-stacks using a custom-made macro in Fiji, as described (*Moro et al., 2020*). Masks of the neuronal somata and neurites were obtained based on the MAP2 signal. Puncta of DCV cargos (BDNF, IA-2) were detected within the neurite mask based on their intensity and dimensions.

## SDS-PAGE WB analysis

To analyze protein levels, high-density neuronal cultures ($400\times10^3$/well in a six-well plate) were lysed in Laemmli sample buffer. Lysates were heated at 95 °C for 5 min, separated by SDS-PAGE using either home-made tris-glycine or commercially available Mini-PROTEAN TGX Stain-Free gels (Bio-Rad 4568096), and transferred to nitrocellulose membrane (Bio-Rad 1620115) using a wet tank transfer method. For BDNF detection, membranes were subjected to crosslinking by 0.5% (v/v) glutaraldehyde directly after the protein transfer according to the published protocol (*Karey and Sirbasku, 1989*) Membranes were blocked in 5% milk powder dissolved in Tris-buffered saline containing 0.1% Tween® 20 (TBS-T). Membranes were incubated with primary antibodies diluted in TBS-T on a shaking platform overnight at 4 °C. Primary antibodies used in this study and their dilutions are listed in the Key Resources Table. Horseradish peroxidase coupled secondary antibodies were applied at 1:10,000 dilution for 1 hr at room temperature. Chemiluminescence-based detection was performed using SuperSignal West Femto Maximum Sensitivity Substrate (Thermo Scientific 34095) on the Odyssey Fc imaging system (LI-COR Bioscience). Immunosignal of some proteins (IA-2, BDNF) was detected using a more sensitive SuperSignal West Atto Ultimate Sensitivity Substrate (Thermo Scientific A38555). Signal intensities of bands of interest were analyzed using Image Studio Lite Software and normalized to the intensity of a loading control (actin, tubulin, or GAPDH). Actin levels were not affected by the loss of tomosyn, as evidenced by comparing band intensities of actin to total protein stain visualized on Mini-PROTEAN TGX Stain-Free gels.

## Detection of SNARE complexes

Detection of SNARE complexes was performed as described previously (*Hayashi et al., 1994*; *Otto et al., 1997*). The protocol is based on the observation that assembled synaptic SNARE complexes (consisting of syntaxin-1, SNAP25, and VAMP2) are preserved in SDS-containing lysis buffer ('SDS-resistant') but sensitive to temperatures higher than 60 °C. To detect syntaxin-1 in SNARE complexes, high-density neuronal cultures (400 K/well in a six-well plate) were lysed in 1% SDS-containing Laemmli sample buffer, passed through an insulin syringe and split into two parts. The first part of the lysates was heated at 95 °C for 5 min (for the detection of total syntaxin-1), while the second part was heated at 37 °C for 5 min (for the detection of syntaxin-1 in SNARE complexes). Lysates were separated by SDS-PAGE using 4–20% gradient gels (Mini-PROTEAN TGX Stain-Free gels, Biorad #4568096). Protein transfer and detection were performed as described above.

## Proteome analysis by mass spectrometry

DIV12-13 hippocampal neurons grown in high-density cultures ($400\times10^3$/well in a six-well plate) were washed once with pre-warmed PBS and lysed in Laemmli sample buffer (50 μl/well). Lysates were passed through an insulin syringe and stored at –80 °C until further processing.

In-gel digestion with Trypsin/Lys-C Mix solution (Promega) was performed as previously described (*Li, 2019*). Peptides were analyzed by micro LC-MS/MS using a TripleTOF 5600 mass spectrometer (Sciex, Framingham, MA, USA). The peptides were fractionated with a linear gradient of acetonitrile using a 200 mm Alltima C18 column (300 μm i.d., 3 μm particle size) on an Ultimate 3000 LC system (Dionex, Thermo Scientific, Waltham, MA, USA). Data-independent acquisition was used with Sequential Window Acquisition of all THeoretical mass spectra (SWATH) windows of 8 Da, as previously described (*Gonzalez-Lozano et al., 2021*).

SWATH data was analyzed using DIA-NN (v1.8) (*Demichev et al., 2020*). A spectral library was generated from the complete mouse proteome with a precursor m/z range of 430–790. Data was searched with 20 ppm mass accuracy, match-between-runs algorithm (MBR) enabled and robust LC as quantification strategy. Propionamide was selected as fixed modification. Downstream analysis was performed using MS-DAP (*Koopmans et al., 2023*; *Koopmans et al., 2022*). The experimental replicate with the lowest number of identified peptides was removed from the analysis (for both genotypes). Only peptides identified and quantified in at least four out of five remaining replicates per genotype were included in the analysis. Normalization was achieved using variance stabilization normalization (Vsn) and mode-between protein methods. The Msqrob algorithm was selected for differential expression analysis, using an FDR-adjusted p-value threshold of 0.01 and log2 fold change of 0.3 to discriminate significantly regulated proteins. All data are available in the PRIDE repository with the identifier PXD038442. Gene ontology enrichment analysis of the differentially expressed proteins was performed using ShinyGO application v0.75 with an FDR-adjusted p-value threshold of 0.05 (http://bioinformatics.sdstate.edu/go/) (*Ge et al., 2020*).

## Electron microscopy

Neurons grown on coated glass coverslips were fixed on DIV14 in 2.5% glutaraldehyde in 0.1 M cacodylate buffer. The cells were postfixed in 1% osmium/1% ruthenium and subsequently dehydrated by increasing ethanol concentrations (30%, 50%, 70%, 90%, 96%, and 100%) before embedding in EPON. After polymerization of the resin for 72 hr at 65 °C, glass coverslips were removed by heating the samples in boiling water. Neuron-rich regions were selected by light microscopy, cut out, and mounted on pre-polymerized EPON cylinders. Ultrathin sections of 70–90 nm were cut on an ultra-microtome (Reichert-Jung, Ultracut E) and collected on formvar-coated single slot grids. Finally, the sections were contrasted with uranyl acetate and lead citrate in an ultra-stainer (Leica EM AC20) and imaged in a JEOL1010 transmission electron microscope (JEOL) at 60 kV while being blinded for the experimental conditions. Synapses, somas, and DCV-rich areas were photographed by a side-mounted Modera camera (EMSIS GmbH). While remaining blinded for experimental conditions, DCV diameters were measured in iTEM software (Olympus), and synapse parameters were quantified in a custom-written software running in Matlab (Mathworks).

## RUSH assay

The protocol is modified from *Boncompain et al., 2012*. Neurons were grown in standard neuronal medium except Neurobasal without phenol red (Gibco 12348017) was used to minimize an auto-fluorescent background during live imaging. On DIV7, half of the medium was refreshed with the same medium lacking the B27 supplement (to reduce extracellular biotin levels). On DIV9, neurons were transfected using the calcium phosphate transfection method with a plasmid encoding a DCV reporter and an ER hook (pCMV-Streptavidin-KDEL-IRES-SBP-EGFP-NPY) together with a 'filler' plasmid (pCMV-mCherry). Neurons were imaged live approximately 16 hr after the transfection upon the addition of 40 µM biotin to the conditioned medium. Time-lapse imaging was performed on a Nikon Ti Eclipse inverted microscope equipped with an A1R confocal module under 40 x oil objective (NA 1.3). LU4A laser unit with a 488 nm wavelength laser was used as the illumination source. Neurons were picked for imaging based on the mCherry fluorescence and were imaged at 37°C and 5% $CO_2$ for 2 hr after the addition of biotin, at a frequency of 1 frame/5 min.

Movies were analyzed in Fiji. Possible lateral drift during recordings was corrected using the Image Stabilizer plugin. Mean intensity of NPY-EGFP within the Golgi mask was measured across the frames and plotted against time in GraphPad Prism. The rate constant of Golgi clearance was determined by fitting a one-phase exponential decay curve to the decay phase of the intensity plot.

For imaging of DCV trafficking in neurites, transfected neurons were imaged in 80 min after the addition of biotin with 0.5 Hz frequency (3 min recordings). Kymographs (space/time plots) were generated from time-lapse videos in Fiji using the KymoResliceWide plugin. Kymographs were analyzed using Kymobutler software (*Jakobs et al., 2019*).

## Quantitative RT-PCR (qRT-PCR)

RNA extraction from neuronal cultures was performed using the ISOLATE II RNA Micro Kit (Meridian Bioscience BIO-52073). cDNA was synthesized from purified RNA using SensiFast cDNA Synthesis Kit

**Table 2.** Summary of statistical analyses applied in this study.

| Figure | Dataset | Groups | n-number* | Statistical test | p-value |
|---|---|---|---|---|---|
| 1C | Cumulative number of DCV fusion events per neuron plotted against time | control<br>DKO | 3 (36)<br>3 (34) | Two-way repeated measures ANOVA | $p_{time \times genotype}$ = 0.4823<br>$p_{time}$ <0.0001 (****)<br>$p_{genotype}$ = 0.3166 |
| 1D | Total number of DCV fusion events/ neuron | control<br>DKO | 3 (36)<br>3 (34) | Mann-Whitney test | p=0.0970 |
| 1E | Number of intracellular NPY-pHluorin puncta/neuron | control<br>DKO | 3 (36)<br>3 (35) | Unpaired t test | p=0.0224 (*) |
| 1F | Fraction of released DCVs (% total pool) | control<br>DKO | 3 (36)<br>3 (34) | Two-way repeated measures ANOVA | $p_{time \times genotype}$ >0.9999<br>$p_{time}$ <0.0001 (****)<br>$p_{genotype}$ = 0.9014 |
| 1G | Fluorescence intensity peaks (averaged per neuron) | control<br>DKO | 3 (36)<br>3 (34) | Mann-Whitney test | p=0.4509 |
| 1H | Band intensities of syntaxin-1 complexes (WB) | DKO (% control) | 4 cultures | One sample t-test (compare to 100%) | $p_{37kDa}$ = 0.2858<br>$p_{80kDa}$ = 0.5714<br>$p_{200 kDa}$ = 0.0002 (***) |
| 1 suppl 1E | Dendrite length/ neuron | control<br>DKO | 3 (30)<br>3 (28) | Unpaired t-test | p=0.5902 |
| 1 suppl 1 F | Number of synaptophysin puncta/ mm dendrite | control<br>DKO | 3 (30)<br>3 (28) | Unpaired t-test | p=0.1380 |
| 1 suppl 1 G | Intensity of synaptophysin signal in neurites | control<br>DKO | 3 (30)<br>3 (28) | Unpaired t-test | p=0.6561 |
| 1 suppl 3 A | Band intensities of SNAP25 complexes (WB) | DKO (% control) | 3 cultures | One sample t-test (compare to 100%) | $p_{25kDa}$ = 0.7265<br>$p_{80 kDa}$ = 0.8454<br>$p_{200 kDa}$ = 0.0064 (**) |
| 1 suppl 3B | Band intensities of VAMP2 complexes (WB) | DKO (% control) | 3 cultures | One sample t-test (compare to 100%) | $p_{15kDa}$ = 0.0585<br>$p_{80 kDa}$ = 0.4283 |
| 2B | Intensity of NPY-pHluorin (ICC) | control<br>DKO rescue | 3 (30)<br>3 (29)<br>3 (29) | One-way ANOVA<br>Tukey's multiple comparisons post hoc test | p<0.0001 (****)<br>$p_{control vs DKO}$ <0.0001 (****)<br>$p_{DKO vs rescue}$ <0.0001 (****) |
| 2B | Intensity of tomosyn (ICC) | control<br>DKO rescue | 3 (30)<br>3 (30)<br>3 (30) | One-way ANOVA<br>Tukey's multiple comparisons post hoc test | p<0.0001 (****)<br>$p_{control vs DKO}$ <0.0001 (****)<br>$p_{DKO vs rescue}$ <0.0001 (****) |
| 2B | Intensity of BDNF (ICC) | control<br>DKO rescue | 3 (20)<br>3 (20)<br>3 (20) | Kruskal-Wallis test<br>Dunn's multiple comparisons test | p=0.0009 (***)<br>$p_{control vs DKO}$ = 0.0009 (***)<br>$p_{DKO vs rescue}$ <0.0253 (*) |
| 2D | Intensity of tomosyn (ICC) | control<br>DKO rescue | 3 (20)<br>3 (20)<br>3 (20) | One-way ANOVA<br>Tukey's multiple comparisons post hoc test | p<0.0001 (****)<br>$p_{control vs DKO}$ = 0.0012 (**)<br>$p_{DKO vs rescue}$ <0.0001 (****) |
| 2F | Band intensities of BDNF (WB) | DKO (% control) | 4 (8) | One sample t-test (compare to 100%) | p<0.0001 (****) |
| 2H | Band intensities of BDNF (WB rescue) | DKO<br>DKO+FL<br>DKO+ΔSNARE | 2 (4) | One sample t-test (compare each condition to 100%) | $p_{DKO}$ = 0.0005 (***) $p_{DKO+FL}$ = 0.1224<br>$p_{DKO+\Delta SNARE}$ = 0.0767 |
| 2I | Band intensities of BDNF (WB leupeptin) | control ±leupeptin<br>DKO ±leupeptin | 2 (4) | Two-way repeated measures ANOVA | $p_{treatment}$ = 0.0640<br>$p_{genotype}$ = 0.0240 (*) |
| 2 suppl 1B | Intensity of EGFP expressed under control of synapsin promoter | control<br>DKO | 3 (30)<br>3 (28) | Unpaired t-test | p=0.0898 |

*Table 2 continued on next page*

*Table 2 continued*

| Figure | Dataset | Groups | n-number* | Statistical test | *p*-value |
|---|---|---|---|---|---|
| | Intensity of IA-2 (neurites) | control<br>DKO | 3 (24)<br>3 (24) | Unpaired t-test | p<0.0001 (****) |
| 2 suppl 2B | Intensity of IA-2 (soma) | control<br>DKO | 3 (24)<br>3 (24) | Unpaired t-test | p=0.0472 (*) |
| 2 suppl 2 C | Band intensities of IA-2 (WB) | DKO<br>(% control) | 3 (9) | One sample t-test (compare to 100%) | p<0.0001 (****) |
| 2 suppl 3B | Intensity of LAMP1 | control<br>DKO | 2 (10) | Unpaired t-test | *p*=0.9847 |
| | Band intensities of LAMP1 (WB) | DKO<br>(% control) | 3 (9) | One sample t-test (compare to 100%) | *p*=0.3716 |
| 2 suppl 3D | Band intensities of LIMP2 (WB) | DKO<br>(% control) | 3 (9) | One sample t-test (compare to 100%) | p=0.7225 |
| 4B | DCV diameter | control<br>DKO | 3 (135 DCVs)<br>3 (64 DCVs) | Unpaired t-test on means from three independent cultures | p=0.0129 (*) |
| 4E | SV diameter | control<br>DKO | 3 (621 SVs)<br>3 (617 SVs) | Unpaired t-test on means from three independent cultures | p=0.2234 |
| 4F | Number of SV/ synapse | control<br>DKO | 3 (149 synapses)<br>3 (147 synapses) | Unpaired t-test on means from three independent cultures | p=0.4713 |
| 4G | Length of active zone | control<br>DKO | 3 (151 synapses)<br>3 (148 synapses) | Unpaired t-test on means from three independent cultures | p=0.4366 |
| 4H | Length of PSD | control<br>DKO | 3 (152 synapses)<br>3 (147 synapses) | Unpaired t-test on means from three independent cultures | p=0.3752 |
| | TGN area | control<br>DKO | 3 (40)<br>3 (39) | Unpaired t-test | p=0.0001 (***) |
| 5B | Nucleus area | control<br>DKO | 3 (40)<br>3 (39) | Unpaired t-test | p=0.1847 |
| 5F | NPY-EGFP intensity in the Golgi plotted against time | control<br>DKO | 3 (33)<br>3 (30) | Two-way repeated measures ANOVA | $p_{\text{time x genotype}}$ = 0.0014 (**)<br>$p_{\text{time}}$ <0.0001 (****)<br>$p_{\text{genotype}}$ = 0.3280 |
| 5G | Time required to reach maximum NPY-EGFP intensity in the Golgi | control<br>DKO | 3 (34)<br>3 (28) | Unpaired t-test | p=0.0444 (*) |
| 5H | Rate constants of NPY-EGFP export from the Golgi | control<br>DKO | 3 (34)<br>3 (30) | Unpaired t-test | p=0.0276 (*) |
| | % anterograde DCVs | control<br>DKO | 2 (12)<br>2 (12) | Unpaired t-test | p=0.9488 |
| 5J | % retrograde DCVs | control<br>DKO | 2 (12)<br>2 (12) | Unpaired t-test | p=0.7938 |
| 5K | Speed of anterogradely trafficking DCVs | control<br>DKO | 2 (11)<br>2 (12) | Unpaired t-test | p=0.8107 |
| 6A | mRNA expression of *Bdnf* transcript isoforms | DKO (logFC relative to control) | 8 cultures | One sample t-test (compare to log$_2$FC = 0) | $p_{\text{total}}$ = 0.0003 (***)<br>$p_{\text{tr1}}$=0.0006 (***)<br>$p_{\text{tr2}}$=0.0032 (**)<br>$p_{\text{tr4}}$ <0.0001 (****)<br>$p_{\text{tr6}}$=0.1183<br>$p_{\text{tr9}}$=0.0012 (**) |

*Table 2 continued on next page*

*Table 2 continued*

| Figure | Dataset | Groups | n-number* | Statistical test | p-value |
|---|---|---|---|---|---|
| 6B | mRNA expression of general DCV cargos | DKO (logFC relative to control) | 8 cultures | One sample t-test (compare to log₂FC = 0) | $p_{Vgf}$ Vgf 0.0006 (***)<br>$p_{Pcsk1}$ Pcsk10.0006 (***)<br>$p_{Pam}$ Pam 0.0032 (**)<br>$p_{Ptprn}$ Ptprn 0.0187 (*)<br>$p_{Scg2}$ Scg20.0177 (*) |
| 6C | mRNA expression of general DCV cargos | DKO (logFC relative to control) | 8 cultures | One sample t-test (compare to log₂FC = 0) | $p_{Npy}$ Npy 0.0002 (***)<br>$p_{Cck}$ Cck 0.0016 (**)<br>$p_{Grp}$ Grp 0.0003 (***)<br>$p_{Sst}$ Sst 0.6986<br>$p_{Tac1}$ Tac10.0201 (*)<br>$p_{Pnoc}$ Pnoc 0.0023 (**)<br>$p_{Vip}$ Vip 0.7972<br>$p_{Pdyn}$ Pdyn 0.8905<br>$p_{Penk}$ Penk 0.4408<br>$p_{Cort}$ Cort 0.0267 (*) |
| 6D | mRNA expression of DCV cargos upon tomosyn re-expression | DKO<br>DKO +tom | 3 cultures | Paired t-test on log₂FC | $p_{Bdnf-tr1}$ Bdnf-tr10.5453<br>$p_{Bdnf-tr4}$ Bdnf-tr40.4470<br>$p_{Vgf}$ = 0.6284<br>$p_{Pcsk1}$ Pcsk10.4625<br>$p_{Grp}$ Grp 0.2974 |
| 6E | Effect of Cre on DCV cargo mRNA expression in *Stxbp5/5l*ˡᵒˣ and WT neurons | ΔCre - *Stxbp5/5l*ˡᵒˣ<br>Cre - *Stxbp5/5l*ˡᵒˣ | 3 cultures | Paired t-test on log₂FC | $p_{Bdnf-tr1}$ Bdnf-tr10.0274 (*)<br>$p_{Vgf}$ = 0.0115 (*)<br>$p_{Pcsk1}$ Pcsk10.0039 (**)<br>$p_{Grp}$ Grp 0.0062 (**)<br>$p_{Gapdh}$ Gapdh 0.2759 |
| | | ΔCre - WT<br>Cre - WT | 3 cultures | Paired t-test on log₂FC | $p_{Bdnf-tr1}$ Bdnf-tr10.0160 (*)<br>$p_{Vgf}$ = 0.0057 (**)<br>$p_{Pcsk1}$ Pcsk10.0040 (**)<br>$p_{Grp}$ Grp 0.0041 (**)<br>$p_{Gapdh}$ Gapdh 0.3046 |
| 6G | Band intensities of BDNF (WB) | ΔCre<br>Cre | 3 (3) | Paired t-test | p=0.0476 (*) |
| | Band intensities of tomosyn (WB) | ΔCre<br>Cre | 3 (3) | Paired t-test | p=0.7470 |
| 6 suppl 2B | Intensity of NPY-pHluorin (ICC) | ΔCre<br>Cre | 3 (18) | Unpaired t-test | p=0.8022 |
| | Intensity of tomosyn (ICC) | ΔCre<br>Cre | 3 (18) | Unpaired t-test | p=0.7105 |

*n-number is the number of independent culture preparations. The number of individual observations (e.g. neurons or lanes) is provided in brackets.

(Meridian Bioscience BIO-65054). qRT-PCRs were run on a QuantStudio 5 system (ThermoFisher Scientific) using SensiFast SYBR Lo-ROX Kit (Meridian Bioscience BIO-94020). qRT-PCR primers (listed in *Table 1*) were validated for the efficiency by running qRT-PCRs on tenfold serial dilutions of cDNA, and for the specificity estimated from melt curves. Only primers with an efficiency of 90–105% were used for experiments. Fold changes (FC) in gene expression were determined using the cycle threshold (CT) comparative method ($2^{-ddCT}$) using *Gapdh* as a housekeeping gene. *Gapdh* CT values were not affected by the genotype or treatment applied. Statistical analysis of the data was performed on log-transformed data (logFC, ddCT).

## Statistical analyses

Sample size was chosen based on the previously published findings from similar experiments using similar measuring techniques. Statistical analyses were performed using GraphPad Prism software. Data distribution was tested for normality using the D'Agostino-Pearson test. Possible outliers were identified using the ROUT method. An F test was used to compare variances between the groups. Statistical tests were chosen based on the distribution of the data sets, as well as on the number of groups being compared. Statistical tests used in individual experiments are specified in

the corresponding figure legends. Full statistical information, including exact p-values, is provided in *Table 2*.

## Materials availability

All DNA constructs and mouse models used in this study are available upon request. Requests should be directed to the lead contact, Matthijs Verhage (m.verhage@vu.nl).

## Acknowledgements

We are indebted to D Schut, L Laan, I Saarlos, R Zalm, J Wortel, J Hoetjes, R Dekker, and I Paliukhovich for their excellent technical assistance. We also thank F Koopmans for his expert advice in proteomics dataset analysis. This work was supported by the DFG (German Research Foundation) postdoctoral fellowship to A Subkhangulova (DFG project number SU 1131/1–1).

## Additional information

### Funding

| Funder | Grant reference number | Author |
| --- | --- | --- |
| Deutsche Forschungsgemeinschaft | SU 1131/1-1 | Aygul Subkhangulova |

The funders had no role in study design, data collection and interpretation, or the decision to submit the work for publication.

### Author contributions

Aygul Subkhangulova, Conceptualization, Data curation, Formal analysis, Funding acquisition, Investigation, Visualization, Methodology, Writing - original draft, Writing – review and editing; Miguel A Gonzalez-Lozano, Formal analysis, Investigation, Visualization, Methodology, Writing – review and editing; Alexander JA Groffen, Resources, Investigation, Writing – review and editing; Jan RT van Weering, Formal analysis, Investigation, Methodology, Writing – review and editing; August B Smit, Conceptualization, Supervision, Project administration, Writing – review and editing; Ruud F Toonen, Matthijs Verhage, Conceptualization, Supervision, Funding acquisition, Project administration, Writing – review and editing

### Author ORCIDs

Aygul Subkhangulova http://orcid.org/0000-0001-8843-0678
Miguel A Gonzalez-Lozano https://orcid.org/0000-0002-7837-151X
Alexander JA Groffen https://orcid.org/0000-0003-0046-4027
Ruud F Toonen https://orcid.org/0000-0002-9900-4233
Matthijs Verhage http://orcid.org/0000-0002-6085-7503

### Ethics

Mice were housed and used for experiments according to institutional guidelines and Dutch laws in accordance with European Communities Council Directive (2010/63/EU).

### Decision letter and Author response

Decision letter https://doi.org/10.7554/eLife.85561.sa1
Author response https://doi.org/10.7554/eLife.85561.sa2

## Additional files

### Supplementary files

• Supplementary file 1. Functional characterization of mitochondrial proteins increased in Double knockout (DKO) proteome.

• MDAR checklist

## Data availability

Proteomics data have been deposited in PRIDE repository with the identifier PXD038442.

The following dataset was generated:

| Author(s) | Year | Dataset title | Dataset URL | Database and Identifier |
|---|---|---|---|---|
| Gonzalez-Lozano MA | 2022 | Effect of the combined loss of tomosyn (STXBP5) and tomosyn-2 (STXBP5L) on the proteome of primary mouse hippocampal neurons | https://www.ebi.ac.uk/pride/archive/projects/PXD038442 | PRIDE, PXD038442 |

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
