## [Editor Report]

This study examines the function of Tomosyn, in dense core vesicle fusion using CRE-mediated deletion in neuronal cultures from mice expressing conditional alleles of tomosyn and tomosyn-2. Tomosyn is a large soluble SNARE protein, where earlier work in multiple species suggested that it functions as a competitive inhibitor of cognate SNARE interactions impairing fusion. The authors show that while loss of tomosyns did not affect dense core vesicle exocytosis, it reduced the expression of several key dense core cargos, including BDNF, with limited impact on intracellular vesicle trafficking or Golgi function. These results suggest that tomosyns impact neuropeptide and neurotrophin secretion by regulating dense core vesicle cargo production but not exocytosis.

---

## [Decision Letter]

**Decision letter after peer review:**

Thank you for submitting your article "SNARE protein tomosyn regulates dense core vesicle composition but not exocytosis in mammalian neurons" for consideration by *eLife*. Your article has been reviewed by 3 peer reviewers, and the evaluation has been overseen by a Reviewing Editor and Gary Westbrook as the Senior Editor. The following individual involved in the review of your submission has agreed to reveal their identity: Silvio O Rizzoli (Reviewer #2).

Essential revisions:

Overall, there is an overall need for better characterization of the KO phenotype. Moreover, some form of "rescue" experiments are needed to validate the specificity of the effects. Ideally, rescue experiments aimed at separating the effects of the SNARE motif versus the WD40 domains of tomosyn should be performed. These experiments should address many of the concerns below.

1. BDNF transcripts have been shown to be regulated by subtle changes in activity or even in response to changes in the balance of excitatory versus inhibitory spontaneous neurotransmission (PMID: 34348149; PMID: 35947950). Therefore, it is important to provide further evidence against an indirect effect of tomosyn loss-of-function beyond CREB phosphorylation.

2. It is also important to go a step further and demonstrate whether these effects require tomosyns' SNARE motif or other regions such as the WD40 domains via rescue experiments. Rescue experiments can also address the specificity of the effects as indicated in #1 above.

3. in Figure 5, supplement 1, the authors document the colocalization of tomosyn with the trans-Golgi network. This analysis should be paralleled with an analysis of the colocalization of this molecule with dense-core vesicle markers and synaptic vesicle markers. The authors' work suggests that the colocalization with vesicular proteins would be poorer than that with the Golgi apparatus. Please analyze carefully.

4. The mouse model developed is well documented and provides a powerful approach to understanding the mechanisms through which tomosyns exert the many effects attributed to them in a wide variety of systems. The hippocampal culture systems utilized in this study allowed the authors to identify many fascinating differences between neurons that develop in the presence or absence of tomosyns. The culture systems used vary greatly in cell density, the ratio of neurons to glial cells, and plating substrates. Given that neuronal development is affected by each of these parameters, it cannot be assumed that the mass spectrometry data obtained for dense cultures accurately describes a single neuron grown on a glial microisland. Was an attempt made to select what appeared to be pyramidal neurons when analyzing single cells? Or were Cre-recombinase-positive cells examined in an unbiased manner? What percentage of neurons were transduced by the Cre-recombinase encoding lentiviruses?

5. Single granule monitoring using exogenous NPY-pHluorin as a DCV reporter allowed the authors to address many aspects of regulated secretion. Although no change in stimulated secretion was observed, a decrease in the total number of DCVs was detected and reduced levels of NPY-pHluorin were observed in the axons and dendrites of TTX-silenced DKO cultures. Increased intracellular degradation of NPY-pHluorin or increased basal secretion of this DCV-reporter could account for the observed decrease in content. Referring to this effect as "regulation" does not seem wise. Please comment.

6. The differences between WT and DKO cultures detected by mass spectrometry need extensive analysis. What could the dramatic increase in mitochondrial proteins mean? Why was it impossible to detect BDNF in DKO cultures by mass spectrometry when immunocytochemistry and mRNA analysis revealed only a two-fold drop? What does the observed increase in glutamatergic and GABAergic markers predict for neuropeptide expression? Certain neuropeptides are more highly expressed in one cell type than in the other. Were the changes in protein expression more prominent for neuronal proteins or glial proteins?

7. Tomosyns are known to affect the exocytosis of a wide variety of vesicles in other systems (e.g. dense core vesicles in β-cells, GLUT4-containing endocytic vesicles in adipocytes and constitutive vesicles in PC12 cells) and may have similarly broad effects in hippocampal neurons. The authors conclude that tomosyns do not regulate the release of dense core vesicle cargo by hippocampal neurons. Using an elegant assay to detect the release of single dense core vesicles containing NPY-pHluorin, a decrease in DCV diameter is detected (and confirmed using electron microscopy). When electrically stimulated over a 25 sec time period, both WT and DKO neurons released 8% of their DCV content (Figure 1E). It is known that DCVs in axons and dendrites respond to different stimuli and that DCV content proteins are released under basal conditions (in the absence of any secretagogue). Please comment.

[Editors' note: further revisions were suggested prior to acceptance, as described below.]

Thank you for resubmitting your work entitled "Tomosyn affects dense core vesicle composition but not exocytosis in mammalian neurons" for further consideration by *eLife*. Your revised article has been evaluated by Lu Chen (Senior Editor) and a Reviewing Editor.

The manuscript has been improved and all reviewers agree that revisions are substantive. However, there are a few remaining issues that need to be addressed/amended in the text as pointed out by Reviewer #3, as outlined below:

*Reviewer #3 (Recommendations for the authors):*

The revised manuscript presents the authors' data in a more logical manner, making it substantially easier to read and understand than the earlier version. The finding that expression of active Cre recombinase in control neurons results in diminished expression of mRNAs encoding Bdnf, Vgf, Pcsk1 and Grp complicates interpretation of the data, but the rescue experiments added to the revised manuscript give the authors a means of dealing with this unexpected complexity. The authors were very responsive to the previous critiques.

I would like to suggest two modifications to the revised text, in order to make it more comprehensible to a general audience. First, cells depend on an array of vesicles to facilitate trafficking and communication between their various membranous organelles. The authors distinguish synaptic vesicles (SVs) from dense core vesicles (DCVs), but often revert to talking about vesicles, with no indication of the type of vesicle to which they are referring. In the Abstract (line 4), the authors refer to "vesicle exocytosis" – here this means SVs, not DCVs, but the text needs to say so. Again, in the Abstract (line 17), they suggest a role for tomosyns in "vesicle biogenesis"; here they are referring to DCVs, and need to be specific. The biogenesis of SVs differs greatly from the biogenesis of DCVs. In the Introduction (line 74), the reader is told that rat sympathetic neurons show a decrease in neurotransmitter release upon depletion of tomosyn – is this a reference to SVs or a reference to DCVs, which contain a variety of peptides used for signaling by sympathetic neurons?

Secondly, the literature makes it abundantly clear that secretion is a very cell-type specific phenomenon. Multiple isoforms of v-SNAREs and t-SNARES play important roles in different tissues, a complexity not really acknowledged in this manuscript. While the first phase of insulin secretion by β-cells depends on Syntaxin 1 and SNAP25, sustained release involves Syntaxin 4 and SNAP23. Which v-SNAREs and t-SNAREs are relevant to the secretion of peptide-containing DCVs in neurons? Platelet α-granules, referred to in the Discussion as an example of DCVs, clearly require multivesicular bodies for their maturation; they receive membrane proteins from the plasma membrane, via the endocytic pathway.

Additional Points:

In setting up their study, the authors state the both SVs and DCVs require an identical basic machinery consisting of sntaxin-1, SNAP25 and VAMP2. While they mention that some of the "many other proteins" involved in the fusion process are "differentially required for SV and DCV exocytosis", they fail to mention the many other ways in which SVs and DCVs (along with myriad of other vesicles in cells) differ – ER/Golgi synthesis and loading of granules vs. local synthesis and vesicle reloading at nerve terminals and types of stimuli required for release, amongst many other differences.

The authors used leupeptin to inhibit lysosomal proteolysis and observed a two-fold increase in BDNF levels in both the double knockout cultures and in wildtype cultures. This is a striking result, unlike what is typically seen in endocrine cells; although it does not explain why the DKO and WT neurons differ, it suggests much more rapid degradation of DCV cargo in neurons than in endocrine cells. The ability of another lysosomal inhibitor, along with quantification of levels of another cargo protein (IA-2, for example) would strengthen this conclusion.

Line 121 – NPY loading per vesicle was not affected by the loss of tomosyns. How do the authors interpret this observation in light of their EM analysis (lines 210-211), which indicates that DCVs are smaller in the tomosyn DKO neurons than in Control neurons (Figure 4C)?

Figure 2 G and I – Panel I shows a greater than two-fold increase in BDNF upon leupeptin treatment of control cultures, but does not show an increase in BDNF in DKO cultures treated with leupeptin. It is difficult to make sense of Panel I and the text that refers to it (lines 177-178).

Figure 5 – Supplement 1C-F – Tomosyn/IA-2 co-localization appears to be much less extensive than Tomosyn/SYPH co-localization. IA-2 is expected to be in DCVs and in vesicles of the endocytic pathway (based on trafficking studies in β-cells).

Line 326 – The authors try to explain "the reluctance of DCV to fuse", suggesting that a poor "choice" has been made. An alternate view to consider is that the difference in ca^2+^-sensitivity exhibited by SVs and DCVs facilitates frequency-dependent responses, with DCVs and their often peptidergic content, recruited only when firing rates are high.

Line 340 – The authors discuss the fascinating α-granule phenotype observed in platelets lacking tomosyn. However, the reader needs to be reminded that α-granules differ in many ways from the DCVs being studied in neurons; membrane associated receptors such as P-selectin are major α-granule components, along with VWF and fibrinogen. The biogenesis/maturation of α-granules is complex (with roles for multivesicular bodies/late endosomes) and quite distinct from that of the DCVs of neurons and β-cells.

Line 342 – In discussing yeast tomosyn orthologs, the authors need to explain that yeast lack both SVs and DCVs and that these yeast orthologs lack a SNARE domain and affect Golgi trafficking, as observed in this study.

---

## [Author Response]

Essential revisions:Overall, there is an overall need for better characterization of the KO phenotype. Moreover, some form of "rescue" experiments are needed to validate the specificity of the effects. Ideally, rescue experiments aimed at separating the effects of the SNARE motif versus the WD40 domains of tomosyn should be performed. These experiments should address many of the concerns below.

In the revised manuscript, we expand the analysis of the DKO phenotype, as requested, and provide results of the suggested rescue experiments. These experiments validate the specificity of most of the effects described in our original manuscript and reveal an additional direct effect of Cre-expression. We also tested the necessity of the SNARE domain of tomosyn in restoration of DCV cargo levels. We believe that the revised version of the manuscript provides a thorough delineation of the roles of tomosyn in neuronal DCV biology. Additionally, our observation on neuronal Cre effects is a finding of interest to many neuroscientists using Cre-expressing neurons as a model system.

1. BDNF transcripts have been shown to be regulated by subtle changes in activity or even in response to changes in the balance of excitatory versus inhibitory spontaneous neurotransmission (PMID: 34348149; PMID: 35947950). Therefore, it is important to provide further evidence against an indirect effect of tomosyn loss-of-function beyond CREB phosphorylation.

We agree and thank the reviewers for these suggestions. To this end, we performed three sets of experiments to test specificity of tomosyn's effect on BDNF levels. First, we tested whether expression of tomosyn restores BDNF protein levels in DKO neurons. Indeed, BDNF levels were restored to 80% of control upon re-expression of tomosyn, as evidenced by ICC (Figure 2C-D), indicating that the decrease in BDNF is largely caused by tomosyn deficiency.

Second, we tested if altered synaptic transmission contributes to the decreased transcription of *Bdnf* and other DCV cargo in DKO neurons (Author response image 1). Blocking action potential-mediated firing for 6 h using tetrodotoxin (TTX) did not rescue the decreased expression of *Bdnf* and other DCV cargos, indicating that the attenuated expression was not caused by altered neuronal activity. Conversely, blocking inhibitory input using picrotoxin (PTX) for 6 h resulted in a massive increase of *Bdnf* (transcripts 1 and 4) mRNA expression. This massive increase blunted the difference between DKO and control neurons, but we find this effect difficult to interpret given the fact that the PTX effect is much larger than the effect of Cre and a ceiling effect may explain the blunted difference. PTX treatment did not affect expression of some other tested cargos (*Grp* and *Npy*) and did not rescue the genotype specific-difference in the mRNA of these genes. As pointed out by the reviewer, PTX may affect *Bdnf* transcription not only by increasing activity, but also via blocking spontaneous inhibitory input at the absence of firing. We tested this contribution by applying both PTX and TTX. We observed no rescue of the decreased *Bdnf* mRNA expression by PTX when action potential-evoked firing was blocked. These data suggest that potential differences in spontaneous inhibitory input do not explain the observed decrease in *Bdnf* mRNA in Cre-expressing *Stxbp5/5l*^lox^ neurons.

Finally, we tested if re-expression of tomosyn rescues transcription of *Bdnf* and other DCV cargos in DKO. Unexpectedly, it was not the case (Figure 6D). Following this unexpected finding, we tested if expression of Cre recombinase *per se* accounts for the decrease in *Bdnf* mRNA in DKO. To do that, we tested effects of Cre expression in wild-type neurons (not bearing lox*P* sites) (Figure 6E). Cre expression in wild-type neurons resulted in an approximately two-fold decrease of *Bdnf* transcript 1 mRNA levels, similar to the decrease observed in *Stxbp5/5l*^lox^ ('floxed' neurons). These data indicate that Cre expression has an unexpected effect on expression of (at least) some *Bdnf* transcript isoforms. However, despite this effect, Cre-expressing wild-type neurons show only 26% decrease in BDNF protein compared to >50% decrease in floxed neurons (Figure 6F-G, 2E-F), indicating that loss of tomosyns specifically affects BDNF protein levels (which can be largely rescued by expression of tomosyn, see the first paragraph). In line with this, total *Bdnf* mRNA expression (tested with primers recognizing all eleven transcript isoforms) was decreased by 34% in DKO (data added to Figure 6A), insufficient to explain the massive decrease of BDNF at the protein level.

We included data on the direct effects of Cre on the transcription of *Bdnf* and other DCV cargos to Figure 6 (panel E) since we think that this finding can be of interest to many neuroscientists using Cre-expressing primary neurons as a model system.

**Author response image 1. sa2fig1:** Effects of tetrodotoxin (TTX) and picrotoxin (PTX) on the mRNA levels of DCV cargos in (Δ)Cre-expressing neurons. (A) mRNA levels of DCV cargo genes upon treatment of (Δ)Cre-expressing cultures with either 1 μM TTX or 50 μM PTX. (B) mRNA levels of DCV cargo genes upon treatment of (Δ)Cre-expressing cultures with a combination of 1 μM TTX and 50 μM PTX. Data are shown as fold changes (FC) relative to the averaged ΔCre-untreated control levels. n=3 culture preparations/ genotype. Dots represent individual cultures; bars are geometric means. Log2FC were analyzed using two-way ANOVA with Holm-Šídák's multiple comparisons test. **p* < 0.05; ***p* < 0.01; ****p* < 0.001.

2. It is also important to go a step further and demonstrate whether these effects require tomosyns' SNARE motif or other regions such as the WD40 domains via rescue experiments. Rescue experiments can also address the specificity of the effects as indicated in #1 above.

We addressed this question by re-expressing either wild-type tomosyn (Stxbp5 isoform m, aa 1-1116) or a mutant lacking the SNARE-domain (ΔSNARE, aa 1-1047) in DKO neurons and monitoring BDNF protein levels by WB (Figure 2G-H). Re-expression of wild-type tomosyn raised BDNF protein levels from 45% to approximately 70% of control levels, and the mutant lacking the SNARE domain had a similar effect. These data suggest that tomosyn regulates BDNF protein levels, and that the SNARE-domain is dispensable for this effect.

3. in Figure 5, supplement 1, the authors document the colocalization of tomosyn with the trans-Golgi network. This analysis should be paralleled with an analysis of the colocalization of this molecule with dense-core vesicle markers and synaptic vesicle markers. The authors' work suggests that the colocalization with vesicular proteins would be poorer than that with the Golgi apparatus. Please analyze carefully.

We complemented our data, as requested, on the colocalization of tomosyn with the TGN marker by performing co-stainings of tomosyn with markers specific for SVs and DCVs (synaptophysin and IA-2, respectively) (Figure 5 —figure supplement 1). We observed a clear colocalization of tomosyn with synaptic and DCV markers, in line with the previously published data (Geerts *et al.,* 2017).

4. The mouse model developed is well documented and provides a powerful approach to understanding the mechanisms through which tomosyns exert the many effects attributed to them in a wide variety of systems. The hippocampal culture systems utilized in this study allowed the authors to identify many fascinating differences between neurons that develop in the presence or absence of tomosyns. The culture systems used vary greatly in cell density, the ratio of neurons to glial cells, and plating substrates. Given that neuronal development is affected by each of these parameters, it cannot be assumed that the mass spectrometry data obtained for dense cultures accurately describes a single neuron grown on a glial microisland. Was an attempt made to select what appeared to be pyramidal neurons when analyzing single cells? Or were Cre-recombinase-positive cells examined in an unbiased manner? What percentage of neurons were transduced by the Cre-recombinase encoding lentiviruses?

In all our experiments, (Δ)Cre-positive cells were analyzed in an unbiased manner. With any new (Δ)Cre-encoding lentivirus batch, we determined a virus titer enabling us to achieve a nearly 100% transduction efficiency (see an example staining in Author response image 2). The efficiency of tomosyn loss was also confirmed by WB (Figure 1 —figure supplement 1B), where no tomosyn band was detected in lysates of Cre-transduced neurons. Thus, no bias was introduced by picking Cre-positive cells.

We agree that a culture system may have an impact on tomosyn null phenotypes and that mass spectrometry data obtained for dense cultures may not accurately describe proteome changes of single neurons grown on glial microislands. Specific culture systems were chosen based on experimental needs (performing mass spectrometry on single neurons on microislands is not feasible). However, the effect of tomosyn's loss on DCV cargo abundance is independent of culture conditions. Similar effects were observed in three different systems: single neurons grown on glial microislands (Figure 1E, 2C-D), sparse network cultures grown on glial feeder layer (Figure 2A-B), and dense cultures on coated plastic (Figure 2E-F). Taken together, these data indicate that the effect of tomosyn on DCV cargo abundance is cell-autonomous and not a consequence of specialized culture conditions.

**Author response image 2. sa2fig2:** Immunostaining of DIV14 network cultures transduced with (Δ)Cre-mCherry at DIV0. Expression of Cre, but not ΔCre lacking the catalytic domain, resulted in a loss of specific tomosyn staining in nearly all imaged neurons. Scale bar 100 μM.

5. Single granule monitoring using exogenous NPY-pHluorin as a DCV reporter allowed the authors to address many aspects of regulated secretion. Although no change in stimulated secretion was observed, a decrease in the total number of DCVs was detected and reduced levels of NPY-pHluorin were observed in the axons and dendrites of TTX-silenced DKO cultures. Increased intracellular degradation of NPY-pHluorin or increased basal secretion of this DCV-reporter could account for the observed decrease in content. Referring to this effect as "regulation" does not seem wise. Please comment.

Thank you for raising this concern. Based on the data on tomosyn orthologs in yeast, Sro7/77, it is indeed plausible that increased intracellular degradation is responsible for the decrease in DCV cargo content in DKO. Loss of Sro7/77 results in mis-targeting of a secretory cargo to vacuole for degradation (Wadskog *et al.,* 2006). In the revised manuscript, we tested whether inhibition of lysosomal degradation rescues the difference in BDNF levels between the genotypes, which was not the case (new Figure 2G, 2I). These data suggest that increased lysosomal degradation does not account for the decrease in BDNF in the absence of tomosyns.

As for the use of the term “regulation”: we replaced it in the revised version of the manuscript by more specific description of the observed effects. We also replaced “regulates” by “affects” in the title of the manuscript.

6. The differences between WT and DKO cultures detected by mass spectrometry need extensive analysis. What could the dramatic increase in mitochondrial proteins mean?

We agree with the need for a better analysis and discussion of the mass spectrometry data, and we now revised this part of the manuscript. We annotated top-upregulated mitochondrial proteins using MitoCarta3.0, an open-source inventory of mitochondrial proteins (Rath *et al.,* 2021) (Supplementary File 1). Interestingly, most of the upregulated proteins were components of the electron transport chain responsible for oxidative phosphorylation (OXPHOS). Other mitochondrial proteins, for example proteins involved in mitochondrial DNA replication and transcription (POLG2, TFAM, EXOG), were not affected in DKO, suggesting normal numbers of mitochondria in DKO. We have added this information to the revised manuscript (lines 190-195). Aerobic oxidation in neurons is tightly linked to neuronal activity and calcium signaling, since neuronal firing is an energy consuming process (Kann *et al.,* 2011; Wong-Riley, 2012; Harris *et al.,* 2012). Tomosyn is a well-known regulator of synaptic activity in both cultured neurons and in the brain, therefore, we speculate that the loss of tomosyns affects.

Why was it impossible to detect BDNF in DKO cultures by mass spectrometry when immunocytochemistry and mRNA analysis revealed only a two-fold drop?

We agree that this discrepancy requires more analyses. We checked our raw mass spectrometry data and found that BDNF was, in fact, detected in DKO, but only in one out of six biological replicates (=independent cultures) and with one peptide only (=one detection event). In contrast, several unique peptides of BDNF were detected multiple times across six biological replicates in the control condition. So technically, BDNF was detected in DKO (once), but this single detection event was not sufficient to statistically quantify the difference between the genotypes. We changed the text in the Results section accordingly (lines 185-188). The single detection of BDNF in one DKO sample rules out any technical issues in our mass spectrometry pipeline (i.e. the detection was possible). We conclude that low levels of BDNF in most DKO samples fall below the detection threshold of our proteomics setup. To validate this conclusion, we have now performed analysis of BDNF in mass cultures by WB, which we added to Figure 2 (panels E-F). DKO samples showed a 65% decrease in BDNF levels by WB. Hence, the revised manuscript now shows a strong reduction in BDNF protein levels, using three independent methods (ICC, WB, mass spectrometry). The size effect is different between the ICC and WB possibly due to the different culture conditions: ICC was performed on single neurons grown on glia microislands, while WB was done on mass cultures grown on plastic.

What does the observed increase in glutamatergic and GABAergic markers predict for neuropeptide expression? Certain neuropeptides are more highly expressed in one cell type than in the other.

We agree, neuropeptide expression is cell type specific and changes in glutamatergic and GABAergic markers could in principle indicate a biased representation of a neuronal population that went into proteomics or Western blot analysis. However, based on the published single-cell transcriptomics data (Saunders *et al.,* 2018), expression of the affected neuropeptides was not confined to a specific neuronal population (e. g. glutamatergic or GABAergic neurons) (lines 266-268 in the Results section). For example, *Grp,* decreased in Cre^+^ neurons, is expressed by a subpopulation of glutamatergic neurons only, while expression of *Npy* (also decreased in Cre^+^ neurons) is restricted to a subpopulation of GABAergic interneurons. Furthermore, expression of *Sst*, which is co-expressed together with *Npy* in a subpopulation of GABAergic neurons, was not affected in Cre^+^ neurons. Thus, changed expression of specific neuropeptides cannot be explained by altered composition of neuronal cultures in Cre^+^ condition.

Were the changes in protein expression more prominent for neuronal proteins or glial proteins?

Given our main aim (testing the role of tomosyns in neuronal DCV exocytosis), our analyses focused on neurons. Neuronal cultures prepared from early postnatal brains do not contain many glia cells as confirmed by MAP2 immunostaining (see Author response image 2). In line with this, we did not detect many canonical glia markers, such as CLDN10, FGFR3, ALDH1L1, in control and DKO lysates by mass spectrometry. Among glial proteins that were detected in our proteome analysis (such as LCAT, ACSBG1, GFAP), none showed altered expression in DKO neurons (Figure 3 source data 1). Thus, though our experimental setup does not allow us to draw strong conclusions on effects of tomosyns on glial proteome, the data suggest that the loss of tomosyns does not result in major changes in glial protein expression.

7. Tomosyns are known to affect the exocytosis of a wide variety of vesicles in other systems (e.g. dense core vesicles in β-cells, GLUT4-containing endocytic vesicles in adipocytes and constitutive vesicles in PC12 cells) and may have similarly broad effects in hippocampal neurons. The authors conclude that tomosyns do not regulate the release of dense core vesicle cargo by hippocampal neurons. Using an elegant assay to detect the release of single dense core vesicles containing NPY-pHluorin, a decrease in DCV diameter is detected (and confirmed using electron microscopy). When electrically stimulated over a 25 sec time period, both WT and DKO neurons released 8% of their DCV content (Figure 1E). It is known that DCVs in axons and dendrites respond to different stimuli and that DCV content proteins are released under basal conditions (in the absence of any secretagogue). Please comment.

A previous study from our lab showed that approximately 80% of DCV fusion events happen in axons of hippocampal neurons (Persoon *et al.,* 2018). The same study also concluded that approximately 6% of the total DCV pool undergoes exocytosis during strong electric stimulation, which is consistent with the results of the current study. We have added these statements to the revised manuscript (lines 110-111, 117-120).

Indeed, DCV exocytosis may occur in the absence of stimulation. However, we observed virtually no fusion events in the first 30 sec of recordings (baseline) or in the period after stimulation (26.5 sec) in neurons of both genotypes, suggesting that spontaneous fusion events are rare events, insufficient to explain the substantial reduction in DCV cargo in DKO neurons. We have added this statement to the revised manuscript (lines 111-113).

References

Boquist L and Lorentzon R (1979) Stereological study of endoplasmic reticulum, golgi complex and secretory granules in the B-cells of normal and alloxan-treated mice. *Virchows Archiv B Cell Pathol* 31: 235–241

Clermont Y, Xia L, Rambourg A, Turner JD and Hermo L (1993) Structure of the Golgi apparatus in stimulated and nonstimulated acinar cells of mammary glands of the rat. *Anat Rec* 237: 308–317

Geerts CJ, Mancini R, Chen N, Koopmans FTW, Li KW, Smit AB, Weering JRT van, Verhage M and Groffen AJA (2017) Tomosyn associates with secretory vesicles in neurons through its N- and C-terminal domains. *PLOS ONE* 12: e0180912

Gurel PS, Hatch AL and Higgs HN (2014) Connecting the Cytoskeleton to the Endoplasmic Reticulum and Golgi. *Current Biology* 24: R660–R672

Harris JJ, Jolivet R and Attwell D (2012) Synaptic Energy Use and Supply. *Neuron* 75: 762–777

Holleran EA and Holzbaur ELF (1998) Speculating about spectrin: new insights into the Golgi-associated cytoskeleton. *Trends in Cell Biology* 8: 26–29

Kann O, Huchzermeyer C, Kovács R, Wirtz S and Schuelke M (2011) Γ oscillations in the hippocampus require high complex I gene expression and strong functional performance of mitochondria. *Brain* 134: 345–358

Müsch A, Cohen D and Rodriguez-Boulan E (1997) Myosin II Is Involved in the Production of Constitutive Transport Vesicles from the TGN. *J Cell Biol* 138: 291–306

Persoon CM, Moro A, Nassal JP, Farina M, Broeke JH, Arora S, Dominguez N, van Weering JR, Toonen RF and Verhage M (2018) Pool size estimations for dense‐core vesicles in mammalian CNS neurons. *The EMBO Journal*: 18

Rath S, Sharma R, Gupta R, Ast T, Chan C, Durham TJ, Goodman RP, Grabarek Z, Haas ME, Hung WHW, *et al.* (2021) MitoCarta3.0: an updated mitochondrial proteome now with sub-organelle localization and pathway annotations. *Nucleic Acids Research* 49: D1541–D1547

Sakisaka T, Yamamoto Y, Mochida S, Nakamura M, Nishikawa K, Ishizaki H, Okamoto-Tanaka M, Miyoshi J, Fujiyoshi Y, Manabe T, *et al.* (2008) Dual inhibition of SNARE complex formation by tomosyn ensures controlled neurotransmitter release. *The Journal of Cell Biology* 183: 323–337

Salcedo-Sicilia L, Granell S, Jovic M, Sicart A, Mato E, Johannes L, Balla T and Egea G (2013) βIII Spectrin Regulates the Structural Integrity and the Secretory Protein Transport of the Golgi Complex*. *Journal of Biological Chemistry* 288: 2157–2166

Saunders A, Macosko EZ, Wysoker A, Goldman M, Krienen FM, de Rivera H, Bien E, Baum M, Bortolin L, Wang S, *et al.* (2018) Molecular Diversity and Specializations among the Cells of the Adult Mouse Brain. *Cell* 174: 1015-1030.e16

Sauvola CW, Akbergenova Y, Cunningham KL, Aponte-Santiago NA and Littleton JT (2021) The decoy SNARE Tomosyn sets tonic versus phasic release properties and is required for homeostatic synaptic plasticity. *eLife* 10: e72841

Wadskog I, Forsmark A, Rossi G, Konopka C, Öyen M, Goksör M, Ronne H, Brennwald P and Adler L (2006) The Yeast Tumor Suppressor Homologue Sro7p Is Required for Targeting of the Sodium Pumping ATPase to the Cell Surface. *MBoC* 17: 4988–5003

Wong-Riley MTT (2012) Bigenomic Regulation of Cytochrome c Oxidase in Neurons and the Tight Coupling Between Neuronal Activity and Energy Metabolism. In *Mitochondrial Oxidative Phosphorylation: Nuclear-Encoded Genes, Enzyme Regulation, and Pathophysiology*, Kadenbach B (ed) pp 283–304. New York, NY: Springer

[Editors' note: further revisions were suggested prior to acceptance, as described below.]

The manuscript has been improved and all reviewers agree that revisions are substantive. However, there are a few remaining issues that need to be addressed/amended in the text as pointed out by Reviewer #3, as outlined below:Reviewer #3 (Recommendations for the authors):The revised manuscript presents the authors' data in a more logical manner, making it substantially easier to read and understand than the earlier version. The finding that expression of active Cre recombinase in control neurons results in diminished expression of mRNAs encoding Bdnf, Vgf, Pcsk1 and Grp complicates interpretation of the data, but the rescue experiments added to the revised manuscript give the authors a means of dealing with this unexpected complexity. The authors were very responsive to the previous critiques.I would like to suggest two modifications to the revised text, in order to make it more comprehensible to a general audience. First, cells depend on an array of vesicles to facilitate trafficking and communication between their various membranous organelles. The authors distinguish synaptic vesicles (SVs) from dense core vesicles (DCVs), but often revert to talking about vesicles, with no indication of the type of vesicle to which they are referring. In the Abstract (line 4), the authors refer to "vesicle exocytosis" – here this means SVs, not DCVs, but the text needs to say so. Again, in the Abstract (line 17), they suggest a role for tomosyns in "vesicle biogenesis"; here they are referring to DCVs, and need to be specific. The biogenesis of SVs differs greatly from the biogenesis of DCVs. In the Introduction (line 74), the reader is told that rat sympathetic neurons show a decrease in neurotransmitter release upon depletion of tomosyn – is this a reference to SVs or a reference to DCVs, which contain a variety of peptides used for signaling by sympathetic neurons?

We thank the reviewer for this suggestion and agree that, especially in the context of our manuscript, the type of vesicle should be clearly indicated throughout the manuscript. We corrected the abstract and main text accordingly.

Secondly, the literature makes it abundantly clear that secretion is a very cell-type specific phenomenon. Multiple isoforms of v-SNAREs and t-SNARES play important roles in different tissues, a complexity not really acknowledged in this manuscript. While the first phase of insulin secretion by β-cells depends on Syntaxin 1 and SNAP25, sustained release involves Syntaxin 4 and SNAP23. Which v-SNAREs and t-SNAREs are relevant to the secretion of peptide-containing DCVs in neurons? Platelet α-granules, referred to in the Discussion as an example of DCVs, clearly require multivesicular bodies for their maturation; they receive membrane proteins from the plasma membrane, via the endocytic pathway.

A large body of evidence suggests that SNAP25 and VAMP2 are strictly required for DCV fusion in mature hippocampal and cortical neurons (Shimojo et al., 2015; Arora et al., 2017; Hoogstraaten et al., 2020). Conversely, VAMP1/3 and SNAP23/29 were dispensable for DCV exocytosis in these neuronal populations, and SNAP47 showed a modulatory effect on BDNF secretion but could not restore secretion in SNAP25-lacking cortical neurons (Shimojo et al., 2015; Hoogstraaten et al., 2020). We added corresponding references to the introduction section (lines 45-47). Though some sensory neurons may require distinct sets of SNAREs for DCV exocytosis (Meng et al., 2007), such neuron type-specific differences in DCV fusion machinery have not been systematically addressed yet. Even less is known about trafficking pathways of transmembrane proteins to DCVs in neurons. Intriguingly, evidence from endocrine cells suggests that transmembrane DCV cargos use endocytic route for delivery to DCVs (Bäck et al., 2010; Hummer et al., 2019), which blurs the line between DCVs and lysosome-related organelles, such as α-granules in platelets (Ma et al., 2021). Nevertheless, as the reviewer pointed out, α-granules are clearly distinct from DCVs in many parameters. We added this important comment to the discussion (lines 357-360).

Additional Points:In setting up their study, the authors state the both SVs and DCVs require an identical basic machinery consisting of sntaxin-1, SNAP25 and VAMP2. While they mention that some of the "many other proteins" involved in the fusion process are "differentially required for SV and DCV exocytosis", they fail to mention the many other ways in which SVs and DCVs (along with myriad of other vesicles in cells) differ – ER/Golgi synthesis and loading of granules vs. local synthesis and vesicle reloading at nerve terminals and types of stimuli required for release, amongst many other differences.

We agree and modified the introduction accordingly (lines 38-43).

The authors used leupeptin to inhibit lysosomal proteolysis and observed a two-fold increase in BDNF levels in both the double knockout cultures and in wildtype cultures. This is a striking result, unlike what is typically seen in endocrine cells; although it does not explain why the DKO and WT neurons differ, it suggests much more rapid degradation of DCV cargo in neurons than in endocrine cells. The ability of another lysosomal inhibitor, along with quantification of levels of another cargo protein (IA-2, for example) would strengthen this conclusion.

We agree that the observed increase in BDNF upon 24h leupeptin treatment is impressive, although it has been reported for cortical neurons (Evans et al., 2011). It has been proposed that neurons synthesize BDNF in excess and regulate its levels by lysosomal degradation. We now corroborated this observation by using another lysosomal inhibitor (E64d) in combination with the ICC-based detection of BDNF. Treatment of WT neurons with E64d for 24h resulted in a massive increase of BDNF signal concentrated in organelles positive for the endolysosomal marker LIMP2, suggesting that a significant portion of BDNF is routed to lysosomes in neurons (Author response image 3). We next tested if another neuronal DCV cargo, CHGB, is subjected to such a rapid degradation by lysosomes. Interestingly, treatment of neurons with E64d for 24h failed to raise CHGB levels, as evidenced by WB. These data suggest that the speed of lysosomal degradation differs between neuronal DCV cargos.

**Author response image 3. sa2fig3:** Effect of lysosomal proteolysis inhibition by E64d (30 μM for 24h) on levels of BDNF and CHGB. (A) Representative images of the BDNF immunostaining and quantification of the somatic BDNF signal in control- and E64d-treated neurons. Scale bar 20 μm. Data were analyzed using a two-tailed unpaired *t*-test. n=8 neurons/ condition. ^**^*p* < 0.01. (B) Colocalization of BDNF with the endolysosomal marker LIMP2 in E64d-treated neurons. Scale bar 20 μm. (C) Levels of CHGB as detected by WB in control and E64d-treated neurons. Equal loading was verified by immunodetection of tubulin. Data were analyzed using a two-tailed paired *t*-test. n=4 culture preparations.

Line 121 – NPY loading per vesicle was not affected by the loss of tomosyns. How do the authors interpret this observation in light of their EM analysis (lines 210-211), which indicates that DCVs are smaller in the tomosyn DKO neurons than in Control neurons (Figure 4C)?

We thank the reviewer for this interesting question. It is possible that DCVs of different size have different probability to fuse, which results in the underrepresentation of fusion events involving very small DCVs in DKO in our NPY-pHluorin based assay. For example, smaller DCVs in DKO may be deficient in certain transmembrane cargos necessary for DCV docking and/or exocytosis, which compromises their fusion competence. We added this comment to the discussion (lines 369-373).

Figure 2 G and I – Panel I shows a greater than two-fold increase in BDNF upon leupeptin treatment of control cultures, but does not show an increase in BDNF in DKO cultures treated with leupeptin. It is difficult to make sense of Panel I and the text that refers to it (lines 177-178).

We agree and replaced the graph in Panel 2I with the graph that includes data from vehicle-treated cultures. The new graph allows to estimate the effect of leupeptin treatment in DKO neurons.

Figure 5 – Supplement 1C-F – Tomosyn/IA-2 co-localization appears to be much less extensive than Tomosyn/SYPH co-localization. IA-2 is expected to be in DCVs and in vesicles of the endocytic pathway (based on trafficking studies in β-cells).

Indeed. We added this comment to the Results section (lines 232-235).

Line 326 – The authors try to explain "the reluctance of DCV to fuse", suggesting that a poor "choice" has been made. An alternate view to consider is that the difference in ca^2+^-sensitivity exhibited by SVs and DCVs facilitates frequency-dependent responses, with DCVs and their often peptidergic content, recruited only when firing rates are high.

We edited the discussion accordingly (lines 338-340).

Line 340 – The authors discuss the fascinating α-granule phenotype observed in platelets lacking tomosyn. However, the reader needs to be reminded that α-granules differ in many ways from the DCVs being studied in neurons; membrane associated receptors such as P-selectin are major α-granule components, along with VWF and fibrinogen. The biogenesis/maturation of α-granules is complex (with roles for multivesicular bodies/late endosomes) and quite distinct from that of the DCVs of neurons and β-cells.

We agree and added this important comment to the discussion (lines 357-360).

Line 342 – In discussing yeast tomosyn orthologs, the authors need to explain that yeast lack both SVs and DCVs and that these yeast orthologs lack a SNARE domain and affect Golgi trafficking, as observed in this study.

We agree. We corrected the discussion accordingly (lines 354-360, 398-400).

References

Arora, S., Saarloos, I., Kooistra, R., van de Bospoort, R., Verhage, M., and Toonen, R. F. (2017). SNAP-25 gene family members differentially support secretory vesicle fusion. *Journal of Cell Science*, *130*(11), 1877–1889. https://doi.org/10.1242/jcs.201889

Bäck, N., Rajagopal, C., Mains, R. E., and Eipper, B. A. (2010). Secretory Granule Membrane Protein Recycles through Multivesicular Bodies. *Traffic*, *11*(7), 972–986. https://doi.org/10.1111/j.1600-0854.2010.01066.x

Evans, S. F., Irmady, K., Ostrow, K., Kim, T., Nykjaer, A., Saftig, P., Blobel, C., and Hempstead, B. L. (2011). Neuronal Brain-derived Neurotrophic Factor Is Synthesized in Excess, with Levels Regulated by Sortilin-mediated Trafficking and Lysosomal Degradation. *Journal of Biological Chemistry*, *286*(34), 29556–29567. https://doi.org/10.1074/jbc.M111.219675

Hoogstraaten, R. I., van Keimpema, L., Toonen, R. F., and Verhage, M. (2020). Tetanus insensitive VAMP2 differentially restores synaptic and dense core vesicle fusion in tetanus neurotoxin treated neurons. *Scientific Reports*, *10*(1), Article 1. https://doi.org/10.1038/s41598-020-67988-2

Hummer, B. H., Maslar, D., Gutierrez, M. S., de Leeuw, N. F., and Asensio, C. S. (2019). Differential sorting behavior for soluble and transmembrane cargoes at the trans-Golgi network in endocrine cells. *Molecular Biology of the Cell*, mbc.E19-10-0561. https://doi.org/10.1091/mbc.E19-10-0561

Ma, C.-I. J., Burgess, J., and Brill, J. A. (2021). Maturing secretory granules: Where secretory and endocytic pathways converge. *Advances in Biological Regulation*, *80*, 100807. https://doi.org/10.1016/j.jbior.2021.100807

Meng, J., Wang, J., Lawrence, G., and Dolly, J. O. (2007). Synaptobrevin I mediates exocytosis of CGRP from sensory neurons and inhibition by botulinum toxins reflects their anti-nociceptive potential. *Journal of Cell Science*, *120*(16), 2864–2874. https://doi.org/10.1242/jcs.012211

Shimojo, M., Courchet, J., Pieraut, S., Torabi-Rander, N., Sando, R., Polleux, F., and Maximov, A. (2015). SNAREs Controlling Vesicular Release of BDNF and Development of Callosal Axons. *Cell Reports*, *11*(7), 1054–1066. https://doi.org/10.1016/j.celrep.2015.04.032